

# Intercomparison of nitrous acid (HONO) measurement techniques in a megacity (Beijing)

Leigh R. Crilley[1], Louisa J. Kramer[1], Bin Ouyang[2], Jun Duan[3], Wenqian Zhang[4], Shengrui Tong[4], Maofa Ge[4], Ke Tang[3], Min Qin[3], Pinhua Xie[3], Marvin D. Shaw[5,6], Alastair C. Lewis[5,6] Archit Mehra[7], Thomas J. Bannan[7], Stephen D. Worrall[7a], Michael Priestley[7 b], Asan Bacak[7], Hugh Coe[7], James Allan[7,6] Carl J. Percival[7 c], Olalekan A. M. Popoola[8], Roderic L. Jones[8] and William J. Bloss[1].

[1]School of Geography, Earth and Environmental Science, University of Birmingham, Edgbaston, Birmingham, B15 2TT, UK
[2]Lancaster Environment Centre, Lancaster University, LA1 4YQ, UK
[3]Key Laboratory of Environment Optics and Technology, Anhui Institute of Optics and Fine Mechanics, Chinese Academy of Sciences, Hefei, 230031, China
[4]Beijing National Laboratory for Molecular Sciences (BNLMS), State Key Laboratory for Structural Chemistry of Unstable and Stable Species, CAS Research/Education Center for Excellence in Molecular Sciences, Institute of Chemistry, Chinese Academy of Sciences, Beijing 100190, China
[5]Wolfson Atmospheric Chemistry Laboratories, University of York, Heslington, York, YO10 5DD, UK
[6]National Centre for Atmospheric Science, UK
[7]Centre for Atmospheric Science, School of Earth and Environmental Sciences, University of Manchester, Manchester, M13 9PL, UK
[8] Department of Chemistry, University of Cambridge, Cambridge, CB2 1EW, UK
[a] now at Chemical Engineering and Applied Chemistry, School of Engineering and Applied Sciences, Aston University, Birmingham, B4 7ET, UK
[b] now at: Atmospheric Science, Department of Chemistry and Molecular Biology, University of Gothenburg, Gothenburg, Sweden
[c] now at Jet Propulsion Laboratory, 4800 Oak Grove Drive, Pasadena, CA 91109, USA

*Correspondence to*: William J. Bloss (w.j.bloss@bham.ac.uk)

**Abstract.** Nitrous acid (HONO) is a key determinant of the daytime radical budget in the daytime boundary layer, with quantitative measurement required to understand OH radical abundance. Accurate and precise measurements of HONO are therefore needed; however HONO is a challenging compound to measure in the field, in particular in a chemically complex and highly polluted environment. Here we report an inter-comparison exercise between HONO measurements performed by two wet chemical techniques (the commercially available LOPAP and a custom-built instrument) and two Broadband Cavity Enhanced Absorption Spectrophotometer (BBCEAS) instruments at an urban location in Beijing. In addition, we report a comparison of HONO measurements performed by Time of Flight Chemical Ionization Mass Spectrometer (ToF-CIMS) and Syft Proton Transfer Reaction Mass Spectrometer (PTR-MS) to the more established techniques (wet chemical and BBCEAS). The key finding from the current work was that all instruments agree on the temporal trends / variability in HONO ($r^2 > 0.97$), yet displayed some divergence in absolute concentrations, with the wet chemical methods consistently higher than the BBCEAS systems by between 12 and 39%. We found no evidence for any systematic bias in any of the instruments, with the exception of measurements near instrument detection limits. The causes of the divergence in absolute HONO concentrations were unclear, and may in part have been due to spatial variability, i.e. differences in instrument location / inlet position.



# 1 Introduction

Nitrous acid (HONO) is one of the key daytime sources of radicals in the boundary layer, as it readily undergoes photolysis to form the OH radical, contributing up to 40% of the OH budget in London (Lee et al., 2016). The OH radical is the primary

oxidant in the troposphere that drives chemical processing, principally the oxidation of volatile organic compounds (VOC) that lead to the formation of ozone and secondary organic aerosols. There are a number of known sources of HONO including direct emissions, heterogeneous reactions, homogenous gas-phase reactions, biological processes and surface photolysis (see reviews by (Kleffmann, 2007;Spataro and Ianniello, 2014). Across urban areas, high spatial heterogeneity in HONO concentration can be observed depending on the proximity to emission sources (Crilley et al., 2016;Lee et al., 2013). Despite,

the importance of HONO to the overall radical budget, the contributions of these different sources, particularly in the urban environment, are poorly understood (See e.g. (Lee et al., 2016;Michoud et al., 2014)).

As a result of the significance of HONO to tropospheric photochemistry, accurate and precise concentration measurements are required, but are challenging due to a number of known potential artefacts in the available approaches: Positive artefacts can

occur in inlet lines, as HONO is easily formed through heterogeneous reactions on wet surfaces (Zhou et al., 2003). Furthermore, the highly reactive nature of HONO means that within inlet lines wall interactions could also lead to a negative artefact, unless inert materials are employed (Pinto et al., 2014). Other challenges include interferences from species such as $NO_2$. There are a number of approaches to measure HONO that can be classed as either wet chemical, spectroscopic techniques or off-line methods (Spataro and Ianniello, 2014). Some of the main instrumentation used recently to measure ambient HONO

in the literature include differential optical absorption spectroscopy (DOAS, e.g. (Perner and Platt, 1979)),  wet chemical techniques (e.g. LOng Path Absorption Photometer (LOPAP), (Heland et al., 2001)), broadband cavity enhanced absorption spectroscopy (e.g. (Duan et al., 2018)), soft chemical ionization mass spectrometry (CIMS, e.g. (Veres et al., 2015)), on-line ion chromatography (e.g. (Stutz et al., 2010;Cheng et al., 2013)) and wet denuder (e.g. (Acker et al., 2004)). In order to compare reported measurements across studies, it is necessary to understand how the different approaches/techniques compare relative

to each other, under actual ambient (field) conditions.

There have been a number of studies reporting intercomparisons between HONO instrumentation (e.g. (Stutz et al., 2010;Ródenas et al., 2013;Pinto et al., 2014;Kleffmann et al., 2006)). Generally, HONO measurements by DOAS systems are used as a reference during inter-comparison studies, as optical methods are artefact free with respect to sampling method,

though impurities in the HONO and $NO_2$ reference spectra can affect retrievals (Stutz et al., 2010;Kleffmann et al., 2006). A further complication with using DOAS systems as a reference is that the spatial averaging inherent in the system means that comparison with point measurements may be subject to bias due to spatial heterogeneities in HONO concentrations. Typically,



most in-situ instruments report higher concentrations during the day compared to simultaneous measurements by a DOAS system, thought to be due to the positive interferences in the in-situ techniques (see e.g. (Febo et al., 1996;Appel et al., 1990;Stutz et al., 2010;Spindler et al., 2003)). An exception is the work by Kleffmann et al. (2006), who reported excellent agreement between a LOPAP and DOAS system in both chamber based and field measurements of HONO under both day and night conditions. The reason for the better performance of the LOPAP is the two channel stripping coil employed in the LOPAP successfully corrects for positive artefacts and chemical interferents during measurement, as demonstrated by Kleffmann and Weisen (2008).

Recently, there have been multi-instrumental inter-comparison studies performed in ambient air and in simulation chambers. These include the Formal Intercomparison of Observations of Nitrous Acid (FIONA) project, which involved 18 instruments measuring within the European Photoreactor (EUPHORE) chamber over a wide range of HONO concentrations (Ródenas et al., 2013). While in general, good agreement was observed during the different experiments of FIONA, at high concentrations (>15 ppb) divergence was observed between some instruments possibly due to some instruments experiencing saturation (Ródenas et al., 2013). These high HONO concentrations however are not typical even in highly polluted locations like Beijing. Pinto et al. (2014) described an inter-comparison of field measurements performed in Houston using a number of instruments for measuring HONO. The instruments tested included a long-path DOAS, a number of wet chemical (including a LOPAP), on-line ion-chromatography and a Time of Flight Chemical Ionisation Mass Spectrometer (ToF-CIMS) using iodide as a reagent ion CIMS. Overall, while good agreement between all the instruments was observed in terms of the temporal trends, the absolute concentrations varied. Pinto and co-workers were unable to pinpoint the cause of the disagreement in absolute concentrations but speculated it might have been due to chemical interference in the in-situ techniques, and the effect of heterogeneous surface reactions due to the distance between some inlets.

Here, we report an inter-comparison exercise of co-located wet chemical and broadband cavity enhanced absorption spectrophotometer (BBCEAS) instruments for measuring HONO in an urban location within Beijing. Ambient concentrations of HONO can vary by several orders of magnitude in Beijing, up to 9 ppb during haze events with a typical concentration of 1.44±1.33 ppb (Wang et al., 2017), making it a challenging location for field measurements. In addition, we report a comparison of HONO retrievals by a ToF-CIMS and Syft PTR-MS to the more established techniques (wet chemical and BBCEAS) for measuring HONO. Based on the inter-comparison findings, the factors that may have influenced the measured concentrations are investigated.



# 2 Method

## 2.1 Site description

Measurements formed part of the Air Pollution and Human Health in a Chinese megacity (APHH-Beijing, www.aphh.org.uk)
program and of the 'An integrated study of air pollution processes in Beijing' (AIRPRO) project, which aimed to understand

atmospheric processes affecting air pollution in Beijing. An overview of the APHH-Beijing project is provided in Shi et al.
(2018). Measurements were performed at the Chinese Academy of Sciences' Institute of Atmospheric Physics (IAP) tower
campus, an urban site located near the 4[th] ring road in the northern suburbs of Beijing. There were two field campaigns, the
first took place during Nov-Dec 2016 (referred to as winter) and second during May-June 2017 (referred to as summer).

## 2.2 Instrument descriptions

An overview of all the instruments that measured ground-level HONO at IAP is provided in Table 1. As instruments of the
same type were used in this study, throughout this paper the instruments will be referred to by their institution name, as per
Table 1. A brief description of each instrument follows.

### 2.2.1 University of Birmingham LOPAP

The University of Birmingham operated a Long Path Absorption Photometer (LOPAP-03, QUMA Elektronik & Analytik

GmbH) at IAP, referred to as the LOPAP throughout. The LOPAP is a wet chemical technique and has been described in detail
in Heland et al. (2001) and Kleffmann et al. (2002). Briefly, a stripping coil is used to sample gas-phase HONO into an acidic
solution where it is derivatized into an azo dye. The light absorption of the azo dye, principally at 550 nm (though higher
wavelengths can also be used), is then measured with a spectrometer using an optical path length of 2.4 m. The LOPAP was
operated and calibrated according to the standard procedures described in Kleffmann and Wiesen (2008). The time resolution

of the LOPAP was 5 minutes and baseline measurements were taken at frequent intervals (8hrs). The operationally defined
detection limit ($2\sigma$) of the LOPAP was calculated to be 35 and 5 ppt for winter and summer, respectively and varied due to
changes in purity of reagents and zero air used. The LOPAP was housed within a temperature controlled shipping container
and sampled at a height of 3m above ground level.

### 2.2.2 Institute of Chemistry, Chinese Academy of Sciences wet chemical HONO analyser

ICCAS applied a custom-built instrument, described in detail elsewhere (Hou et al., 2016). It is a wet chemical technique
similar in principle to the LOPAP. Gas-phase HONO is almost completely absorbed by an absorption solution into a two-
channel stripping coil, where it forms an azo dye, detected by absorption spectroscopy at a wavelength of 550 nm with an
optical path length of 0.5 m. The instrument has a detection limit ($2\sigma$) of 134 ppt for a response time of 5 min.





### 2.2.3 University of Cambridge BBCEAS

The University of Cambridge ran a three-channel BBCEAS instrument during the campaign, with one channel measuring $NO_2$ and HONO simultaneously in the UV (362-374 nm) wavelength region. Reference absorption cross-sections of HONO (Stutz et al., 2000) and NO2 (Voigt et al., 2002), were fitted to the absorption coefficient to retrieve HONO and NO2 concentrations

Details of the instrument can be found in Kennedy et al. (2011). Two mirrors (Layertec 109053) with peak reflectivity of ~99.95% at 365 nm were used to form the cavity. Given an inter-mirror distance of 92 cm, the effective absorption pathlength in the case of an empty cavity was around 1.8 km. The limit of detection ($1\sigma$) was determnined to be 25 ppt for a 60 s time response.

Both instrument inlet and the optical cavity were made of perfluoroalkoxy alkane (PFA) which is well known for its chemical inertness. During the winter phase of the campaign, instrument inlet was placed at the top of the container and was about 3 m from ground. During the summer phase however, the instrument was moved to an adjacent container, also housing the AIOFM instrument, and the height of instrument inlet was changed accordingly from ~3 to ~2 m.

To allow a more stable cavity throughput (i.e. to minimise flow turbulence effect on the optical signal), the sampling flow was set to 2 LPM for the HONO/$NO_2$ measurement channel. This was close to the very low end of the operational range of the flow controller (0-50 LPM), leading to the actual flow rate potentially differing from that set. Post-campaign analysis identified that this affected both the dilution factor (dilution of the sample flow by the two mirror purge lines) and the length the sample gas occupied in the cavity. A post-campaign calibration was therefore performed by injecting a known amount of $NO_2$ into the
cavity under otherwise identical operating conditions, and a scaling factor of ~1.27 was found to be necessary to account for these two factors, and was then applied to the measured $NO_2$ and HONO concentrations.

### 2.2.4 Anhui Institute of Optics and Fine Mechanics BBCEAS

The custom-built BBCEAS instrument from the Anhui Institute of Optics and Fine Mechanics (AIOFM), Chinese Academy of Sciences has been described in detail in Duan et al. (2018); therefore only a brief description is given here. Light is emitted
by a single light emitting diode with peak wavelength of 365 nm, FWHM of 13 nm and is introduced into the resonant cavity, consisting of a pair of high reflective (HR) mirrors with reflectivity of about 0.99985 at 368 nm, separated by 55.0 cm. The emergent light intensity passing through the cavity exit mirror is received by an Ocean Optics QE65000 spectrometer through an optical fibre with 600 μm diameter, and a 0.22 numerical aperture.

In order to avoid the drift of the center wavelength of the LED, the temperature of the LED was controlled to be approximately $20 \pm 0.05$ °C by using a TEC unit. In order to prevent particulate matter from entering the cavity and reducing the effect of particulate matter on the effective absorption path, a 1 μm PTFE filter membrane (Tisch Scientific) was used in the front end





of the sampling port. The time resolution of the IBBCEAS instrument was 1 min, and the 2σ detection limit of HONO was about 120 pptv. The fitting wavelength range was selected as 359–387 nm, with the same reference cross-sections used in the retrieval of $NO_2$ and HONO as for the University of Cambridge instrument. Sample loss and secondary formation of HONO were both considered in this instrument and the measurement error of HONO was estimated to be approximately 9%.

**2.2.5 University of Manchester ToF-CIMS**

A time of fight chemical ionisation mass spectrometer (ToF-CIMS) (Lee et al. 2014), using an iodide ionisation system was coupled with a filter inlet for gases and aerosols (FIGAERO) originally developed by Lopez-Hilfiker et al. (2014) and recently described and characterised by Bannan et al. (2019). The detailed setup during this campaign can be found in Zhou et al. (2018). The FIGAERO enabled near simultaneous, real-time measurements of both the gas and particle phase composition.

Only gas phase data is presented here, so of every 75 minutes of continuous data 35 minutes (particle phase mode) are omitted. The gas phase inlet consisted of 5 m ¼" I.D. PFA  tubing connected to a fast inlet pump with a total flow rate of 13 standard litres per minute (slm) from which the ToF-CIMS sub-sampled 2 slm.

Methyl iodide gas mixtures ($CH_3I$) in $N_2$ were made up in the field using a custom-made manifold (Bannan et al. 2014). 20

standard cubic centimetres per minute (sccm) of the $CH_3I$ mixture was diluted in 4 slm $N_2$ and ionised by flowing through a Tofwerk X-ray ionisation source. This flow enters an ion molecule region (IMR) which was maintained at a pressure of 400 mbar using an SSH-112 pump fitted with an Aerodyne pressure control box to account for changes in ambient pressure. A short segmented quadrupole (SSQ) was positioned behind the IMR and was held at a pressure of 2 mbar using a Tri scroll 600 pump.

During the campaign, gas phase backgrounds were established through regularly overflowing the inlet with dry $N_2$ for five continuous minutes every 45 minutes (Priestley et al. 2018) and were applied consecutively. The overflowing of dry $N_2$ will have a small effect on the sensitivity of the instrument for those compounds whose detection is water dependent. Here we find that due to the very low instrumental background for HONO, the absolute error remains small (<33 ppt) and is an acceptable

limitation in order to measure the wide range  of different compounds that can be detected using the iodide CIMS.
Field calibrations were regularly carried out using known concentration formic acid gas mixtures made in the manifold. The instrument was calibrated for a range of other species after the campaign, and relative calibration factors were derived using the measured formic acid sensitivity as has been performed previously  (Le Breton et al., 2014;Le Breton et al., 2017;Bannan et al., 2014;Bannan et al., 2015). HONO was measured at m/z 174 as $I.HNO_2^-$ during the period of 27th May – 17th June 2017.

A stable and pure gas phase source of HONO was generated for calibrations using the method described by Ren et al. (2010) and Febo et al. (1995) and a sensitivity of 0.28 cps/ppt was applied to the data with a LOD of 33 ppt. Data analysis is performed using the "Tofware" package (version 2.5.11) running in the Igor Pro (WaveMetrics, OR, USA) environment. The mass axis was calibrated using $I^-$, $I_2^-$ and $I_3^-$. Extracted high resolution time series were then normalised to the iodide reagent ion trace.



### 2.2.6 University of York Syft PTR-MS

The data presented in this paper were measured using a Voice200 Selected ion flow tube mass spectrometer (SIFT-MS, Syft Technologies, Christchurch, New Zealand). This instrument consists of a switchable reagent ion source capable of rapidly switching between multiple reagent ions. Reagent ions are generated in a microwave discharge ion source region, acting on an

air/water mix at a pressure of approximately 440 mTorr to generate $H_3O^+$, $NO^+$, and $O_2^+$. These ions are extracted into the upstream quadrupole chamber maintained at a pressure of approx. 5 $\times 10^{-4}$ torr, using a 70 L/s turbo-molecular pump. The reagent ions pass through an array of electrostatic lenses and the upstream quadrupole mass filter, and those not rejected by the mass filter are passed into the flow tube where they are carried along in a stream of nitrogen and selectively ionise target analytes. Gas phase data presented herein was determined using the $H_3O^+$ reagent ion only. Sampling was carried out at a

height of approx. 3 m using a gas phase inlet consisting of 3.5 m ¼" I.D. PFA tubing connected to a diaphragm inlet pump (KNF) at a total flowrate of 5 standard litres per minute (slpm), from which the SIFT-MS sampled approximately 2 slpm through an in house built pressure controlled inlet maintaining a consistent absolute inlet pressure of 0.5 bar. The flow tube is pumped by a 35 m³/h scroll-type dry pump (Edwards) resulting in a mass flow controlled gas flow of 25 sccm for the nitrogen carrier gas (research grade, BOC) and a sample flow of 100 sccm taken from the pressure controlled inlet system. These flows

result in a continuous total flow tube pressure of approximately 460 mTorr and a reaction time of approx. 8 ms (Hera et al., 2017). During the campaign, gas phase backgrounds were established through regularly overflowing the sample inlet with dry nitrogen for five continuous minutes every hour.

Nitrous acid (HONO) was measured at product ion m/z 48 ($H_2ONO^+$) following protonation using the $H_3O^+$ reagent ion. As a

HONO calibration source was not available, a relative transmission curve calibration (between 19-107 m/z) was carried out daily  as describe by Taipale and colleagues (Taipale et al., 2008) using a 2 ppm gas standard containing ethylene, Isobutane, benzene, toluene, ethylbenzene, tetraflourobenzene, hexafluorobenzere and octafluoro benzene.

### 2.3 Formal inter-comparison

The formal inter-comparison measurements took place during the 9th-14th November 2016, and comprised the four established

techniques for measuring HONO (2 wet chemical and 2 BBCEAS). All instruments had a sampling height of 3m during the inter-comparison, and inlets were located as close as possible to each other. The BHAM and ICCAS instrument were housed within the same shipping container, with their respective inlet heads located beside each other on the roof. The CAM and AIOFM BBCEAS instruments were housed in separate containers, with inlets located approximately 5 and 10 m, respectively, from the two wet chemical inlet heads. At the completion of the formal winter intercomparison, the inlet locations changed for

some of the instruments.





There was no formal inter-comparison between all four instruments in the summer campaign. The BHAM, CAM and ICCAS inlets were located in the same position as per the winter intercomparison at the start (22nd May - 30[th] June 2017). Therefore, further analysis was performed between these three instruments for this period to examine for any changes in their relationships compared to the winter measurements. The AIOFM instrument was housed within the same container as per the winter however, the inlet was located approximately 3 m further away from other instruments in the summer. On the 30[th] May, the CAM instrument was moved to the same container as the AIOFM, with the inlets located approx. 3 m from each other.

The ToF-CIMS and Syft PTR-MS were not initially set up to measure HONO at IAP, but were able to provide some useful data during the summer measurements and are therefore compared to the more established techniques.

## 2.4 Data Analysis

The BHAM and ICCAS instruments were operated with a time response of 5 min, and as this was the longest (Table 1), 5 min averages were used for all instruments in the inter-comparison analyses. For each instrument, their normal quality control procedures were applied and only data that passed the quality control was used for subsequent analysis. Data analysis was performed in R (v 3.5.1) using the openair package (Carslaw and Ropkins, 2012) and the lmodel2 package for Reduced Major Axis (RMA) regression analysis.

## 3 Results

### 3.1 Winter formal intercomparison

#### 3.1.1 Time series

The time series (Figure 1) demonstrates that while all instruments captured the same temporal trends, the absolute concentrations differed. The correlation coefficients from regression analyses show that there is little scatter between measurements from the different instruments with r values being consistently between 0.96 and 0.98 (Table 2). Overall, the BHAM LOPAP measurements were consistently the highest, followed by ICCAS, AIOFM and CAM. The slopes from the RMA analysis demonstrated that none of the instruments were in agreement (Table 2) within their stated error (Table 1) during the formal inter-comparison exercise. Therefore, in the following sections we investigate possible reasons to account for the lack of agreement between instruments.

#### 3.1.2 Analysis of Co-efficient of Variance

The co-efficient of variance (CV) is defined as the standard deviation divided by the mean and is used to compare the relative degree of variation between datasets. Therefore, the CV is a measure of precision, and as a guide a CV of 0.1 is considered as acceptable by the US EPA for PM instruments (Sousan et al., 2016). From Fig 2, the CV was fairly consistent throughout the winter inter-comparison, at an average of 0.28±0.07. The CV was however observed to increase at the end of the inter-




comparison, coinciding with period of the lowest mean HONO concentration (< 1 ppb, Fig 2) suggesting that as the concentrations approached the detection limit (DL) of some instruments (Table 1), there was a decrease in the level of agreement. $NO_2$ is a potential interferent for the measurement of HONO for both wet chemical and BBCEAS instruments (Heland et al., 2001). For the BBCEAS instruments, the fitting algorithm used should isolate HONO and $NO_2$ very well in

theory; however, the applied reference $NO_2$ spectrum might contain absorption signatures from HONO. This would result in a higher $NO_2$/HONO ratio retrieved from the BBCEAS compared to ambient air, and consequently reporting a low HONO mixing ratio (Kleffmann et al., 2006). However, the Voigt et al (2002) $NO_2$ cross-section, used by both BBCEAS instruments, has previously been shown to have negligible HONO absorption structures (Veitel, 2002). We also note that HONO reference spectrum should contain little structure from $NO_2$. Fig 2, however, demonstrates no apparent relationship between the CV and

$NO_2$. This likely reflects the efforts taken during processing and measurement to reduce the influence of interference from $NO_2$ in all instrument types.

### 3.1.3 Normalised difference analysis

The systematic error for each instrument can be calculated by normalised sequential difference (NSD) according to Eqn 1 (Arnold et al., 2007):

$$NSD = \frac{(Conc_t - Conc_{t+1})}{(Conc_t \times Conc_{t+1})^{0.5}} \qquad (1)$$

The results are shown in Fig S1 (Supporting Information), and as each instrument showed a symmetrical and Gaussian distribution it suggests there was no internal systematic bias for any given instrument. We then examined the normalized difference (ND) between pairs of instruments to explore inter-instrument variability, calculated according to Eqn 2 (Pinto et al., 2014):

$$ND_{ij} = \frac{(c_i - c_j)}{(c_i + c_j)} \qquad (2)$$

where $C_i$ and $C_j$ denote HONO levels measured by any pair of instruments (BHAM, ICCAS, AIOFM or CAM). We also calculated the co-efficient of divergence (CD), which is a normalized measure of the similarity between two measurement time series, derived via Eqn 3 (See Pinto et al. (2014) and references therein):

$$CD_{ij} = \sqrt{\left(\frac{1}{p}\right) \times \Sigma (ND_{ij})^2} \qquad (3)$$

Where p is the number of observations and $ND_{ij}$ is defined in Eqn 2. A CD of one means the time series are completely different, while of CD of zero indicates that they are identical. The calculated CD for each instrument pair is shown in Table 3 and demonstrate that each of the two overall approaches - wet chemical (BHAM and ICCAS) and BBCEAS (AIOFM and CAM) - agreed well internally. The ICCAS and AIOFM also agreed well, but CAM and BHAM had a higher CD with AIOFM and ICCAS (Table 3).





If there is no difference between a pair of instruments, then the calculated ND should be scattered around zero, and from Fig 3 this was not observed for any instrument pair, pointing to differences between instruments. The ND was evaluated as function of wind direction and measured HONO concentration (Fig 3), to explore if ambient concentration or spatial heterogeneity could explain the disagreements. From Figure 3, for all instrument pairs the highest ND, and therefore largest relative

difference, between instruments was at low HONO mixing ratios (ca. <1 ppb) and was also associated with a westerly direction. At high wind speeds, the ND was also high between all instrument pairs (Fig S2, Supporting Information). As we observed high ND at relatively high wind speeds, it would suggest that spatial variability in ambient HONO concentrations did not affect the intercomparison as high wind speeds typically homogenise ambient concentrations from point and local sources. Overall from Figs 3 and S2, the periods of low HONO concentration, high wind speeds and westerly winds all coincided during the

formal winter intercomparison making it difficult to disentangle the influence of these factors on the observed ND.

### 3.1.4 Instrument agreement at low concentrations

There was evidence from the CV (Fig 2) and ND (Fig 3) analyses that the level of agreement between instruments decreased at low HONO mixing ratios. Therefore we applied RMA correlation analysis for periods when the HONO level was below 2 ppb (as measured by CAM), and the results are shown in Table 4. From Table 4, the observed slopes between the BHAM-

ICCAS-AIOFM at low concentrations (<2 ppb) were similar to those for the whole winter inter-comparison dataset (Table 2). This suggests that the difference in measured concentrations between these instruments (as indicated by the slope) was not related to concentration. There was however a notable decrease in the slope for the low concentrations between CAM and the other three instruments compared to whole inter-comparison (Tables 2 and 4, respectively), potentially pointing to changes in the CAM readings at lower concentrations. This change may be related to differences in instrument sensitivity (Table 1).

### 3.2 Summer Measurements

While there was no formal intercomparison during the summer measurements, at the start of the summer measurements the BHAM, ICCAS and CAM instrument inlets were co-located as per the winter formal intercomparison. The relationship between instruments for this period is shown in Table 5. The agreement (gradient) between BHAM and ICCAS improved in the summer to 0.91 compared to winter (0.77) but with slight changes in intercept (Tables 2 and 5). A change was also observed

between CAM and ICCAS with a lower slope observed for the start of summer (Table 5) compared to winter intercomparison period (Table 2).

The AIOFM instrument started measuring halfway through the summer campaign, and while the instrument was housed in the same container the inlet location was a few meters further from the other three instruments than in the winter inter-comparison.

As a result, we compared the instrument readings for one week after the AIOFM instrument started measuring (7[th]-14[th] June 2017), with the results shown in Table 6. From Table 6, the agreement between instruments of the same type were within their stated uncertainties for the summer. However, when comparing the between the two different instrument types (wet chemical



and BBCEAS, e.g. AIOFM-BHAM, etc), the agreement was notably worse compared to the winter (Tables 2 and 6). The exception was that the agreement between the BHAM and CAM, which was similar in the winter and summer (Tables 2 and 5), despite the CAM inlet moving further away from BHAM inlet on the 30[th] May 2017 (Table 6).

Generally, the level of agreement between instruments varied between the summer and winter and this may reflect spatial variability in HONO concentrations as some of the instrument inlet locations varied from summer to winter. In the summer, the CAM inlet moved closer to the AIOFM inlet and the agreement between the two BBCEAS improved to be within uncertainty (Table 6). However, we also note that the BHAM and ICCAS inlets were in the same location winter and summer and yet the agreement between instruments changed considerably between the two measurement periods. We re-calculated the

ND for two inter-comparison periods analysed in the summer (Tables 4 and 5) and found no relationship between the ND and wind direction (Figures S4 and S5, Supporting Information). This suggests that during the summer measurements the wind direction may have exerted less influence on the spatial variability of the HONO levels or that observed relationship between wind direction and ND in winter was associative not causal.

### 3.2.1 Performance of MANC ToF-CIMS

Measurements from the Manchester ToF-CIMS are compared to the BHAM and CAM instruments for the summer campaign as these instruments had the best data coverage for periods when the MANC instrument was measuring, as well as representing the typical upper and lower measurements (Fig 1). In general the MANC instrument captured the temporal trends (r > 0.84) but recorded higher HONO concentrations than the other instruments (Table 7). Similar distributions were observed between the BHAM and MANC datasets, with the exception of a number of outliers for MANC (Figure 4). We note that MANC was

not co-located with either BHAM or CAM instrument and while this will likely have affected the inter-comparison, the results do point to the MANC instrument capturing the temporal trends but at a higher concentration than the other instruments (157-239%, Table 7).

### 3.2.2 Performance of the YORK Syft PTR-MS

The York Syft PTR-MS was primarily used for measuring VOC fluxes and so did not typically measure at ground level. To

enable an inter-comparison with the other techniques, the YORK instrument measured at ground level for an extended single continuous period, from 1800 30/05/2017 until 0900 31/05/2017. The results are shown in Fig 5 and while the short time period and spatial distance between inlets (approx. 50m) limits the conclusions that can be drawn, it is clear that the YORK instrument captured the temporal trends (r of 0.9-0.96 compared to other techniques) and gave comparable concentrations to the BHAM instrument (slope of 0.78). This suggests that the YORK instrument was measuring HONO at reasonable

concentrations. Furthermore, we note that a co-located PTR-MS (PTR-TOR 1000, Ionicon) was unable to see a HONO signal despite both instruments using $H_3O^+$ to detect HONO.





## 4 Discussion

From the literature, the recent inter-comparison of ambient field measurements of HONO concentrations described by Pinto et al. (2014) is the most relevant to the current work. Overall, in their study Pinto and co-workers found that in general the level of disagreement between instruments was greater than the stated uncertainties for each instrument. While there was some evidence for a chemical interference (but Pinto and co-workers could not identify the compounds responsible definitively), there were additional factors that also appeared to affect the inter-comparison. The best agreement in Pinto et al. (2014) was found for instruments with co-located inlets compared to instruments with inlets several meters apart and so points to spatial heterogeneity in HONO concentrations (possibly due a source on the roof surface) affecting the inter-comparison. Overall, the results from the current work are similar to that observed previously (Pinto et al. 2014), as there was a separation of up to 13 meters between some instrument inlets and this may have affected the results for the inter-comparison in the current work. With respect to photolysis, the lifetime of HONO at midday ranged from 17-300 and 9-33 min for winter and summer, respectively (depending upon weather / cloud cover / aerosol loading) and may have contributed to spatial heterogeneity in HONO concentrations.

Duan et al. (2018) presented results of an inter-comparison at a rural site in China between a BBCEAS and a LOPAP, with good agreement observed (slope of 0.94 and $r^2$ 0.89). The slope appeared to deviate from linearity above approximately 2 ppb suggesting that at higher concentrations the relationship was changing, as observed here between CAM and wet chemical techniques (BHAM and ICCAS) (Tables 2 and 4). A divergence in the measured concentrations at high concentrations was also observed for all instruments as part of the FIONA inter-comparison (Ródenas et al., 2013), but at much higher concentrations (>15 ppb) than most of those encountered here. However, we did not observe such a change in relationship at high and low concentrations between the AIOFM-BHAM/ICCAS, suggesting that this result was not related to instrument type. Furthermore, as the measurements from Duan et al. (2018) were performed in a rural site, conditions may also be more homogenous mix compared to an urban location and this may explain why there was better agreement between the LOPAP and BBCEAS in their study compared to the current work.

Throughout this work, the wet chemical techniques generally measured higher concentrations than the spectroscopic techniques, in agreement with previous studies (e.g. Stutz et al., 2010; Pinto et al., 2014). Pinto et al. (2014) suggested the possible cause may be a positive chemical interference in the wet chemical instruments. The observed dependence of the normalized difference between each instrument pair on wind direction may reflect changes in composition affecting the instrument readings. We note that the two-channel stripping coil used in the sampling inlet for both the BHAM and ICCAS instruments should account for any chemical interferences, particularly in the gas-phase (Kleffmann et al., 2002). Particle-phase nitrite is also expected to be corrected by the two-channel system, and we also note that based on the effective Henry's law constant for HONO and the typically acidic nature of Beijing aerosol that little particle-phase nitrite would be expected



(Kleffmann et al. 2006). Therefore, we do not expect there to be significant chemical interference on the BHAM and ICCAS wet chemical instruments.

In the current work, differences were observed between measurements from instruments of the same type (BBCEAS and wet chemical). The cause of this disagreement was difficult to pinpoint, but may reflect differences in calibration and corrections applied by each group. In particular the BHAM and ICCAS instruments inlets were next to each other during the inter-comparison (<0.5m), and thus the differences likely reflect more differences in the operating conditions of the BHAM and ICCAS instruments. Both instruments used the same nitrite standard for calibration. Notably, there is a significant difference in DL between the instruments (Table 1) likely due to the different methodologies for determining baseline correction. For example, the BHAM instrument used zero air sampled at the inlet to determine the baseline, whereas ICCAS used water introduced into the wet chemical side of the instrument. Tests have shown that water results in a lower baseline measurement for the LOPAP (approx. 80-100 ppt), however when comparing the two wet chemical methods this would only result in a change in the intercept, not the slope.

The scaling factor to correct for the discrepancy in flow rate applied to the CAM instrument after the campaigns is unlikely to be the cause of the disagreement between the two BBCEAS. The two BBCEAS systems agree to within ±10% for $NO_2$ measurements, and the larger disagreement for HONO (13%, Table 2) likely reflects higher spatial variability of ambient HONO compared to $NO_2$, as the CAM and AIOFM inlets were the furthest apart during the formal winter inter-comparison. The agreement between the two BBCEAS decreased at lower concentrations and this may reflect differences in DL (Table 1). The two BBCEAS instruments were found to be in better agreement in the summer compared to the winter. This may reflect the inlets being closer in summer compared to the winter, however there was still a distance of 3m between inlets. We do not know the reasons why the agreement between the AIOFM and CAM instruments changed in the summer compared to winter. Whilst there were some variability in path length and purge flows between the two BBCEAS systems, these are not thought to account for the discrepancy as they did not vary winter to summer. Furthermore, another factor may be losses/production of HONO and $NO_2$ on the inlet filter and/or tubing losses/production that may also influence the intercomparison results as these were naturally different across systems due to different residence times (~3 s and ~0.5 s for the CAM and AIOFM BBCEAS, respectively). Laboratory tests using the same tubing material (Teflon) have however shown that neither the production nor the loss of HONO were significant in the CAM instrument (to less than a few percent) even at considerably longer inlet and cavity residence times, suggesting this was insignificant.

Generally, the agreement between instrument pairs varied from winter to summer, with the exception of CAM and BHAM instruments. As all instruments were operated and calibrated according to the same procedures in winter and summer, there were no changes in instrument operation that can explain these changes and as such, the cause is unclear. The concentrations





observed during summer (mean of 1.2±0.9 ppb (1σ)) were typically lower compared to the winter (mean of 2.35±1.9 ppb (1σ)), and this may have affected the results.

## 5 Conclusions

Overall, from the winter inter-comparison all instruments were found to agree on the temporal trends and variability in HONO
(r >0.97), yet displayed some divergence in absolute concentrations (slopes of 0.61-0.88), with the wet chemical methods consistently somewhat higher than the BBCEAS systems. We found no evidence for any systematic bias in any of the instruments, with the exception of measurements near instrument detection limits. There was evidence that the relationship between some instruments varied for the different measurement periods (e.g. winter/summer) however the reason for this change was unclear. When considering the mass spectrometric methods (MANC ToF-CIMS and YORK PTR-MS), these
captured the temporal trends in HONO concentrations but were found to differ in absolute concentration relative to the other instrumentation.

There was no evidence for a definitive cause of systematic bias between the four instruments during the formal HONO inter-comparison, which might justify scaling or excluding results from one or more instruments. As a result, we could not say with
confidence, which instrument (if any) provided the 'correct' measurement of HONO concentration. Therefore, to meet the needs of the wider APHH-Beijing program for a single ground level HONO measurement, a merged HONO dataset was produced using the mean and range concentration of the four instruments that participated in the formal winter inter-comparison (two wet chemical and two BBCEAS). This merged dataset will be used for future ground level analyses (e.g. model evaluation) across the APHH-Beijing program.

**Authors contributions**

Study conceived by BO and LC. Measurements were performed by LC, LK, BO, JD, WZ, MS, AM, TB, SW, JA and AB. Formal analysis performed by LC, LK and BO. All co-authors contributed to data curation. LC prepared the manuscript with contributions from all co-authors.

**Acknowledgements**

This work was funded by the UK Natural Environment Research Council (NERC), Medical Research Council and Natural Science Foundation of China under the framework of Newton Innovation Fund (NE/N007190/1 and NE/N007077/1). WJB, LJK and LRC acknowledge additional support by the UK NERC through the project Sources of Nitrous Acid in the Atmospheric Boundary Layer (SNAABL, NE/M013405/1). Authors also acknowledge the support of staff at Institute of



Atmospheric Physics in Beijing, in particular Prof Pingqing Fu, during the measurement campaigns. Original data are available on request from the authors, and have been deposited in the (open access) CEDA repository, available for public download following the project embargo period.

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



**Table 1: Instrumentation measuring HONO at IAP.**

| Institution | Instrument | Manufacturer | DL (ppt) | Error | Time resolution | Reference |
|---|---|---|---|---|---|---|
| **Birmingham (BHAM)** | Wet Chemical (LOPAP.03) | QUMA | Winter: 35 Summer: 5 (2σ, 30s) | 10% | 5 min | (Heland et al., 2001) |
| **ICCAS** | Wet Chemical | Custom built | 134 (2σ, 30s) | 10% | 5 min | (Hou et al. 2016) |
| **Cambridge (CAM)** | BBCEAS | Custom built | 25 (1σ, 60s)) | 9% | 5 s | (Kennedy et al., 2011) |
| **AIOFM** | BBCEAS | Custom built | 120 (2σ, 60s) | 9% | 1 min | (Duan et al. 2018) |
| **Manchester (MANC)** | ToF-CIMS | Aerodyne Research Inc / Tofwerk. | 33 (2σ, 60s) | 19 % | 1 Hz | (Priestley et al., 2018) |
| **York** | Syft PTR-MS Voice ultra 200 | Syft Technologies | 130 (2σ, 60s) | 22% | 19 secs (1min averaged) | (Hera et al., 2017) |





**Table 2: Results of the reduced major axis regression analysis with 95% confidence intervals during the formal winter inter-comparison. Variability shown is the 95% confidence interval of the slope and intercepts.**

| INSTRUMENTS | INTERCEPT | SLOPE | R | N |
|---|---|---|---|---|
| **BHAM-ICCAS** | 0.09±0.04 | 0.77±0.01 | 0.97 | 865 |
| **BHAM-AIOFM** | -0.18±0.03 | 0.71±0.01 | 0.98 | 1070 |
| **BHAM-CAM** | -0.23±0.03 | 0.61±0.01 | 0.98 | 1125 |
| **ICCAS-AIOFM** | -0.20±0.03 | 0.88±0.01 | 0.98 | 954 |
| **ICCAS-CAM** | -0.38±0.04 | 0.82±0.01 | 0.97 | 991 |
| **AIOFM-CAM** | -0.09±0.03 | 0.87±0.01 | 0.96 | 1206 |



**Table 3: Calculated CD values for each instrument pair during the winter inter-comparison.**

|         | ICCAS | AIOFM | CAM  |
|---------|-------|-------|------|
| **BHAM**  | 0.11  | 0.22  | 0.32 |
| **ICCAS** | -     | 0.12  | 0.21 |
| **AIOFM** | -     | -     | 0.11 |



**Table 4: RMA regression analysis (with 95% confidence intervals) for times when the abundance of HONO was less than 2 ppb as measured by CAM during the formal winter intercomparison period. Variability shown is the 95% confidence interval of the slope and intercepts.**

| INSTRUMENTS | INTERCEPT | SLOPE | R | N |
|---|---|---|---|---|
| BHAM-ICCAS | -0.01±0.06 | 0.82±0.02 | 0.96 | 437 |
| BHAM-AIOFM | -0.25±0.03 | 0.76±0.01 | 0.98 | 529 |
| BHAM-CAM | -0.06±0.03 | 0.53±0.01 | 0.95 | 613 |
| ICCAS-AIOFM | -0.23±0.05 | 0.91±0.03 | 0.95 | 478 |
| ICCAS-CAM | -0.12±0.05 | 0.68±0.02 | 0.91 | 556 |
| AIOFM-CAM | 0.07±0.03 | 0.72±0.02 | 0.92 | 655 |



**Table 5: RMA regression relationships of HONO measured by BHAM-ICCAS-CAM during the co-located measurements at the start of the summer campaign (22$^{nd}$ May-30$^{th}$ May 2017). All three inlet locations were the same as the formal winter intercomparison. Variability shown is the 95% confidence interval of the slope and intercepts.**

| INSTRUMENTS | INTERCEPT | SLOPE | R | N |
|---|---|---|---|---|
| BHAM-ICCAS | -0.24±0.02 | 0.91±0.01 | 0.97 | 2061 |
| BHAM-CAM | -0.34±0.02 | 0.61±0.01 | 0.90 | 1233 |
| ICCAS-CAM | 0.21±0.02 | 0.69±0.01 | 0.85 | 1346 |



**Table 6: RMA regression relationships (with 95% confidence intervals) of HONO measured by all instruments in the middle of the summer campaign (7th-14th June 2017). Note that BHAM and ICCAS inlets were in same location for this period. The CAM instrument had moved to the same container as AIOFM, whose inlets were 3 meters apart. Variability shown is the 95% confidence interval of the slope and intercepts.**

| INSTRUMENTS | INTERCEPT | SLOPE | R | N |
|---|---|---|---|---|
| BHAM-ICCAS | 0.17±0.04 | 0.93±0.03 | 0.90 | 900 |
| BHAM-AIOFM | -0.18±0.03 | 0.61±0.02 | 0.81 | 1377 |
| BHAM-CAM | 0.02±0.03 | 0.65±0.02 | 0.86 | 1395 |
| ICCAS-AIOFM | -0.08±0.02 | 0.43±0.01 | 0.81 | 1153 |
| ICCAS-CAM | 0.07±0.03 | 0.53±0.02 | 0.82 | 1167 |
| AIOFM-CAM | 0.20±0.01 | 1.07±0.02 | 0.92 | 1982 |



**Table 7: RMA regression relationships of HONO measured by BHAM-CAMB-MANC for the whole summer period. Variability shown is the 95% confidence interval of the slope and intercepts.**

| INSTRUMENTS | INTERCEPT | SLOPE | R | N |
|---|---|---|---|---|
| **BHAM-MANC** | -0.35±0.06 | 1.57±0.04 | 0.84 | 1896 |
| **BHAM-CAM** | 0.00±0.02 | 0.63±0.01 | 0.84 | 4106 |
| **CAM-MANC** | -0.30±0.05 | 2.39±0.05 | 0.88 | 2372 |



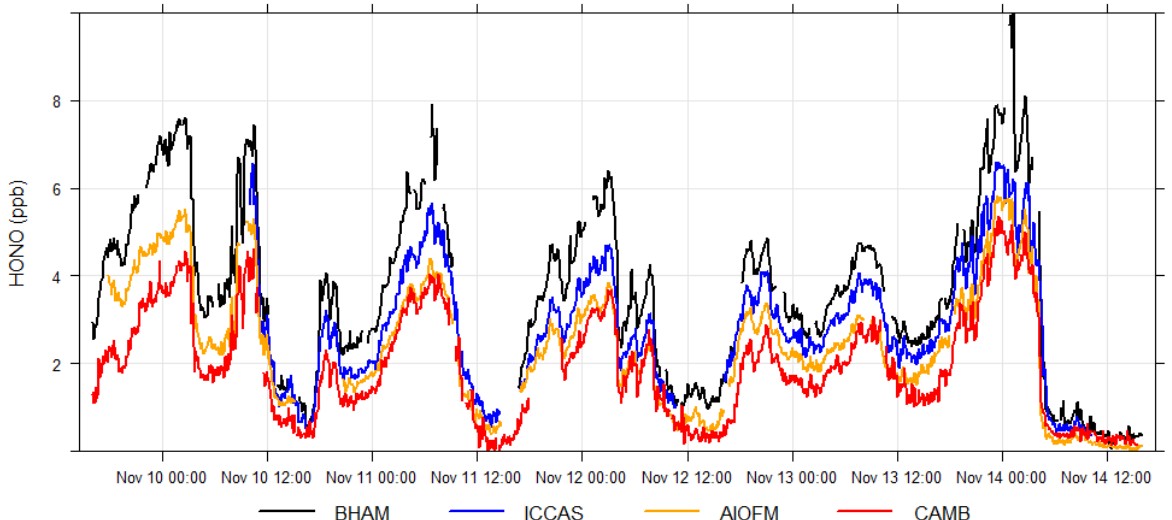

**Figure 1: Time series of the measured mixing ratios during the formal winter inter-comparison for each instrument**




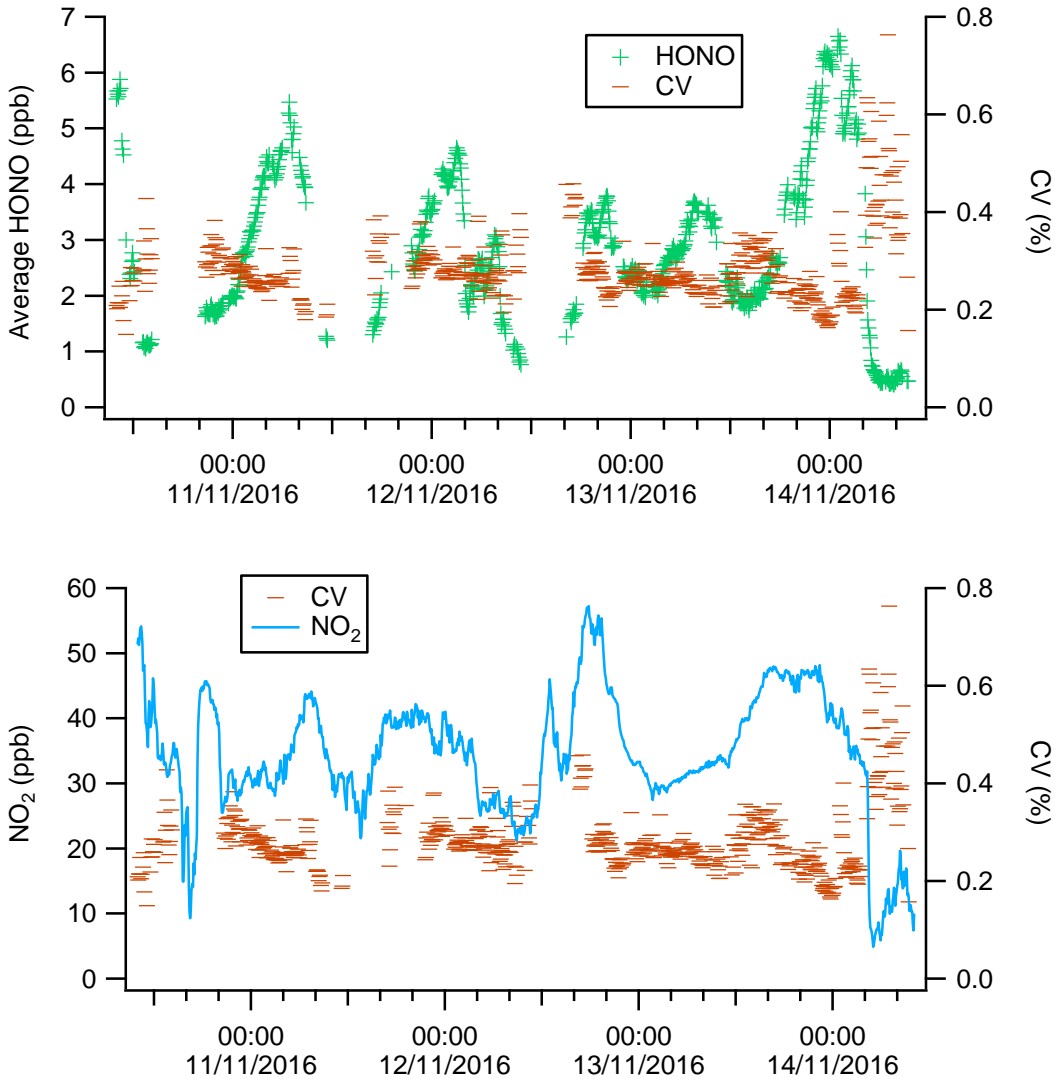

**Figure 2: Time series of the co-efficient of variance (CV), mean mixing ratio of HONO (top) and NO₂ (bottom) during the winter inter-comparison. Note only when all four instruments were measuring were the mean HONO and CV calculated.**





**Figure 3: Normalised Differences (ND) for each instrument pair as a function of wind direction coloured by measured HONO**
5  **concentration (ppb) for the winter intercomparison. Note the different scales for the y-axes and HONO abundance colour key.**





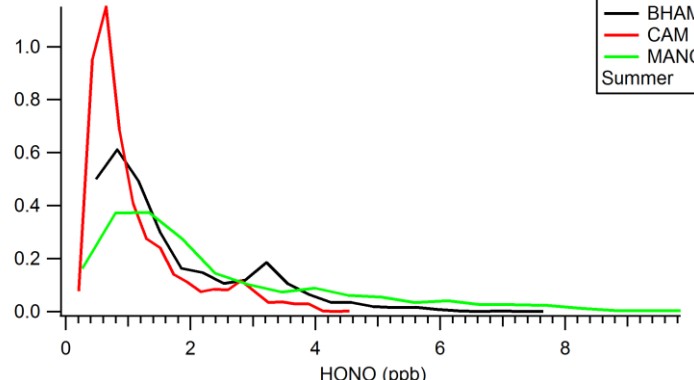

**Figure 4: Histogram of measured summer concentrations (only for periods when all three instruments were measuring).**



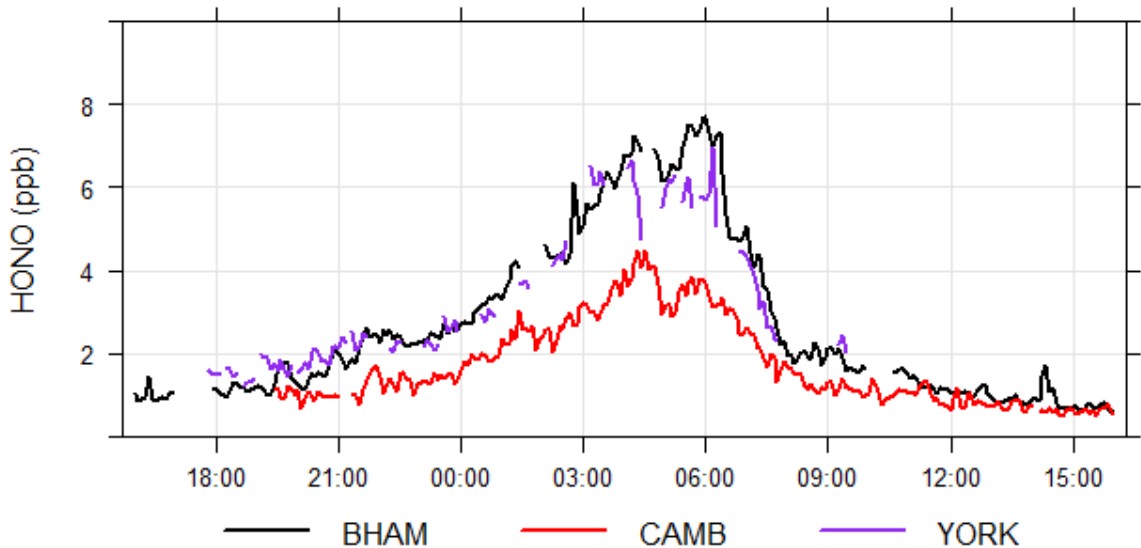

**Figure 5: Time series for the period when the YORK instrument measured at ground level during the summer campaign.**