# Peer review of "Intercomparison of nitrous acid (HONO) measurement techniques in a megacity (Beijing)"

_Atmospheric Measurement Techniques, 2019_

## Short Comment (SC1) · 25 Jul 2019

This paper presents measurements of HONO by several instruments and is of interest to many researchers (myself included) who are interested in HONO. Given the growth in the use of the iodide ToF chemical ionization mass spectrometer method, it would be helpful for further details of the instrument's operation to be presented.

The IMR pressure is held at 400 mbar, which is higher than that used for most research groups' IMRs ($\sim 50 - 100$ mbar). What is the reason for operating at this relatively high pressure?

Backgrounds were determined using dry N2. The authors state "The overflowing of dry N2 will have a small effect on the sensitivity of the instrument for those compounds

whose detection is water dependent. Here we find that due to the very low instrumental background for HONO, the absolute error remains small (<33 ppt)". According to the same Lee et al. 2014 paper referenced in the manuscript, the sensitivity (as configured in Lee et al) varies by a factor of five between dry conditions and the most humid conditions tested (PH2O = 0.8 mbar). It would be useful if there were a figure that showed the time series for m/z 174 for a typical sample & background measurement cycle to support the statement that the uncertainty in the background is essentially inconsequential.

The sensitivity is quoted as 0.28 counts/s/ppt – at what humidity? A graph of the HONO sensitivity as a function of humidity would be helpful, especially since it could differ than that measured by Lee et al given the different IMR pressures used. Is this sensitivity normalized to 1,000,000 cps of reagent ion? What is the total reagent ion signal? Finally, additional details on how potential HONO loss or formation in the sampling lines was investigated for most of the instruments used would be helpful. Thanks.

---

## Referee Comment (RC1) · Anonymous Referee #1 · 14 Aug 2019

This study reported an inter-comparison exercise between HONO measurements performed by multiple instruments. The results show that despite of good agreement on the temporal trends, the wet chemical methods consistently higher than the BBCEAS systems by between 12 and 39%. The reason for the divergence was not clear and the authors have speculated and discussed the potential influence of instrument locations.

HONO is one of the most important precursors of OH radicals. Reliable measurements of HONO are key to understand its origin and role in the atmospheric chemistry. This study provides a great dataset to examine the performance and potential problems of HONO detectors in an atmosphere subject to strong anthropogenic influences. Overall, I think it is a nice study and would recommend its publication with revisions. My main concerns are that (1) the authors may put too much weight on the contri-

bution/discussion of spatial heterogeneity. Do you think that you may see an reverse relation (wet chemical concentration < BBCEAS) if you exchange the sampling locations? (2) the discussion about the daytime difference seems to be missing or mixed in the discussion. The 12% to 39% may not reflect the real difference up to several folds during daytime, the most relevant periods for the photochemical reactions. Please find detailed comments below.

Comments:

Page 1 line 36 "with the wet chemical methods consistently higher than the BBCEAS systems by between 12 and 39%." Is it the case for daytime, night time, I saw 100%?

Page 1 line 38 "The causes of the divergence in absolute HONO concentrations were unclear, and may in part have been due to spatial variability, i.e. differences in instrument location / inlet position." Did you check the concentration of other trace gases aerosols, T, RH, etc?

Page 2 Introduction "contributing up to 40% of the OH budget in London (Lee et al., 2016)." The authors may consider referring to its contribution to the primary production of OH, where you can find more references (e.g., Kleffmann et al., GRL, 2005; Acker et al., GRL, 2006; Su et al., JGR, 2008) including measurements in Beijing (Yang et al., 2014).

Page 2 Introduction "There are a number of known sources of HONO including direct emissions, heterogeneous reactions, homogenous gas-phase reactions, biological processes and surface photolysis (see reviews by (Kleffmann, 2007;Spataro and Ianniello, 2014))." When talking about different sources, I'd suggest crediting the original research rather than exclusive referring to reviews.

Page 2 line 14 "Positive artefacts can occur in inlet lines, as HONO is easily formed through heterogeneous reactions on wet surfaces (Zhou et al., 2003)." The study of Zhou is not about the heterogeneous reactions on inlets. It is discussing the formation

of HONO and NOx from photolysis of HNO3. Concerning the artefacts in inlet lines, you could refer to the studies of Zhou et al. (GRL,2002) and Su et al. (AE, 2008). The LOPAP style instruments have an outdoor sampling unit without additional inlet. Is this the case for the two wet chemical analyzers used here? What's the length of inlet of the other unspecified instruments?

Page 8 line 26 "The co-efficient of variance (CV) is defined as the standard deviation divided by the mean and is used to compare the relative degree of variation between datasets." The authors should explain how to calculate CV explicitly because the mean and deviation can be calculated for an individual dataset. My understanding is that here you calculate CV of data from different instruments at each time step and it is better to clarify this

Page 9 line 5 "the applied reference NO2 spectrum might contain absorption signatures from HONO. This would result in a higher NO2/HONO ratio retrieved from the BBCEAS compared to ambient air, and consequently reporting a low HONO mixing ratio (Kleffmann et al., 2006). However, the Voigt et al (2002) NO2 cross-section, used by both BBCEAS instruments, has previously been shown to have negligible HONO absorption structures (Veitel, 2002)." This part of discussion is not optimal. On one hand, you stated that the applied reference spectrum might be problematic. On the other hand, you referred to Veitel et al. 2002 saying that it is negligible. Actually, this is also the information provided by Kleffmann et al. (2006). Since you were using the one from Voigt et al., it should solve the impurity issue as suggested by Kleffmann et al. (2006). I'd suggest a reformulation of this part.

Page 9 line 15: "The systematic error for each instrument can be calculated by normalised sequential difference (NSD) according to Eqn 1 (Arnold et al., 2007)" The authors should better explain equation 1. What's the meaning of t and t+1, is it time? You later used i, j, what's their difference? Why it reflects a systematic error? People may have different ways to define a systematic error vs a random error. But the fact that BHAM was always higher than CAMB for me is a clear evidence for systematic

errors.

Page 10 line 2 "The ND was evaluated as function of wind direction and measured HONO concentration (Fig 3), to explore if ambient concentration or spatial heterogeneity could explain the disagreements." Do you have other parameters (e.g., SO2, O3, NOx, etc) that have been measured in different containers? A comparison of these parameters may give you a better idea about the spatial heterogeneity, and help to disentangle it from the other effects. Besides, the spatial heterogeneity and HONO concentration might not be the only reasons for the observed differences. Artefacts and intrinsic limitation of individual methods could also results in different kinds of disagreement. These, however, were not discussed in this study. Do you consider these as minor issues here?

Page 10 Section 3.1.4. Here the authors investigated the concentration dependence of instrument agreement through comparison of slopes between the whole dataset and a subset of data. Can you give more details why you were expecting a concentration dependence, or what could lead to a concentration dependence? Do you think the detection limit would be an issue here? This can be easily checked by exclusively comparing data above the threshold. I also have some technical questions: (1) how did you conclude "the observed slopes between the BHAM15 ICCAS-AIOFM at low concentrations (<2 ppb) were similar to those .."? Because according to the confidence interval in Table 2 and 4, these slopes could be significantly different; (2) what's the criteria of choosing 2 ppb as the threshold, will the results change if you take other values, e.g., 0.5 ppb? A more straight forward way to investigate the concentration dependence would be directly plot the difference or normalized difference against the concentration. HONO concentration also has a strong diurnal cycle, with a minimum around noon time with active photochemistry and temperature. The concentration dependence of bias might also be caused by a diurnal cycle of the interference signals as discussed in Su et al. (AE 2008). Can you check if the difference in your study also has a prominent diurnal cycle?

Page 10 line 32: "However, when comparing the between the two different instrument types (wet" Some words missing?

Page 11 line 5 "this may reflect spatial variability in HONO concentrations as some of the instrument inlet locations" I am not convinced about the role of spatial variability. Because how to maintain a large gradient of HONO in 3 meters? Could you make some estimation about the heterogeneity of the source/driving force by taking a typical turbulent diffusion coefficient?

Page 12 line 30 "We note that the two-channel stripping coil used in the sampling inlet for both the BHAM and ICCAS " Did you compare the signal of the second channel of the two-LOPAP style instruments? This channel is supposed to provide information about most inference species. Detailed analysis (e.g., correlation study with different nitrogen-containing compounds, or of it diurnal variation) may give you a better idea of the interference problems. Such a comparison can also be used to check the performance of both instruments and spatial heterogeneity.

Figure 2: The CV has a unit of "%". Is it correct? A CV of 1% is too small according to Fig. 1.

Reference Acker, K., D. Moller, W. Wieprecht, F. X. Meixner, B. Bohn, S. Gilge, C. Plass-Dulmer, and H. Berresheim (2006), Strong daytime production of OH from HNO2 at a rural mountain site, Geophys. Res. Lett., 33(2), L02809, doi:10.1029/2005GL024643.

Kleffmann, J., T. Gavriloaiei, A. Hofzumahaus, F. Holland, R. Koppmann, L. Rupp, E. Schlosser, M. Siese, and A. Wahner (2005), Daytime formation of nitrous acid: A major source of OH radicals in a forest, Geophys. Res. Lett., 32, L05818, doi:10.1029/2005GL022524.

Su, H., Y. F. Cheng, M. Shao, D. F. Gao, Z. Y. Yu, L. M. Zeng, J. Slanina, Y. H. Zhang, and A. Wiedensohler (2008), Nitrous acid (HONO) and its daytime sources at a rural site during the 2004 PRIDE-PRD experiment in China, J. Geophys. Res., 113, D14312,

doi:10.1029/2007JD009060.

Su, H., Y. F. Cheng, P. Cheng, Y. H. Zhang, S. Dong, L. M. Zeng, X. Wang, J. Slanina, M. Shao, and A. Wiedensohler (2008), Observation of nighttime nitrous acid (HONO) formation at a non-urban site during PRIDE-PRD2004 in China, Atmospheric Environment, 42(25), 6219-6232, doi:https://doi.org/10.1016/j.atmosenv.2008.04.006.

Yang, Q., et al. (2014), Daytime HONO formation in the suburban area of the megacity Beijing, China, Science China-Chemistry, 57(7), 1032-1042, doi:10.1007/s11426-013-5044-0.

Zhou, X., He, Y., Huang, G., Thornberry, T.D., Carroll, M.A.,Bertman, S.B. (2002). Photochemical production of nitrous acid on glass sample manifold surface. Geophysical Research Letters 29 (14), 1681.
* * *

---

## Referee Comment (RC2) · Anonymous Referee #2 · 21 Aug 2019

Review of "Intercomparison of nitrous acid (HONO) measurement techniques in a megacity (Beijing)", AMT-2019-139 This paper presents a multi instrument comparison of HONO observations collected in Beijing. Considering the atmospheric importance of HONO, it is very important to determine the accuracy of various measurement techniques. The most refreshing part of this manuscript is that the authors make no attempts to disguise the differences between the various techniques, and do not attempt to identify a true measurement technique. This allows for a neutral consideration of the state of available technology to measure ambient HONO, while making it difficult to understand the cause of the variations. The presentation and discussion could use some work cleaning up and organizing the details of the results as it is sometimes confusing to keep track of what methods or time periods are being compared or can

in fact be compared. I also find that the authors put a lot of stock in the location of sampling points as an explanation for the discrepancies, while providing limited evidence to support this. This work is valuable in that it draws attention to the continuing problem of evaluating the robustness of HONO observations. This work would benefit from attempting to arrive at a point where evidence-based suggestions can be provided to inform future observations or comparison studies. I support the publication of this work contingent upon addition minor revisions as suggested above and in the detailed comments provided below.

Page 2, line 3-4: This is oddly specific towards London considering the number of publications on daytime HONO, and that this manuscript does not address London HONO. Suggest broadening the citations for the generalization of the introduction.

Page 2, line 9: The statement "depending on the proximity to emission sources" is meant to mean what? Emission of HONO, NOx? Direct, indirect?

Page 3, line 14: "highly polluted locations like Beijing" needs a citation or evidence otherwise that Beijing is highly polluted.

Page 4, line 3: edit to read "Measurements were performed as part of"?

Page 4, line 15: "referred to as LOPAP throughout" is a bit confusing since you just introduced that instrument will be referred to by the institution and have already defined LOPAP acronym earlier in the manuscript.

Page 4, line 25: It would be helpful in this work to explicitly discuss what the main differences between the two LOPAP techniques are.

Page 5, line 12: AIOFM is not yet defined in the text.

Page 54, line 16-21: This is pretty hard to understand here. Are you simply stating that the flow controller was not properly calibrated and describing the process to calibrate it? If so this seems like it is an unnecessary discussion as one would expect that you have calibrated your flow controllers to get accurate measurements.

Page 6, line 3-4: How exactly did you consider sample loss and secondary formation of HONO in the instrument? That is not exactly the most straightforward thing to do, and it is glossed over here.

Page 6, line 24 and elsewhere: One major thing missing here that is likely very important is a detailed discussion on how the function instrument zero was determined and the associated errors involved. For example, how was the absolute error of 33ppt calculated here? Is that the variability between instrument zeros? Is once every 45 minutes sufficient to capture the variability in the background for the CIMS. Are the timescales for others important? From figure 1 it seems like incorrect background subtraction for each instrument could easily explain the difference in magnitude between the various instruments, especially considering the high degree of correlation observed. I'd really like to see a more detailed discussion of the background determination and subtraction and its impact on the reported values for each instrument.

Page 6, line 30: The lack of humidity in the dry N2 zero should significantly impact the HONO sensitivity, you should include the humidity dependent calibration of your instrument. Does HCOOH display the same dependence? This could impact your ability to use formic acid as a surrogate to track changes in instrumental sensitivity. While this is commonly done in past studies it may not be accurate. This is also a concern for the SIFT instrument backgrounds.

Page 7, line 20: The compounds used in the relative transmission calibration curve are very difference from HONO. How well is method expected to perform in approximating the HONO sensitivity considering these differences? Are the portions of a given molecules real sensitivity that are not accurately captured by this method, for example surface losses or secondary ion chemistry, or fragmentation?

Page 7, line 24-25: This is a poorly written sentence and needs work.

Page 8, line 1-2: It is very difficult throughout this manuscript to keep track of when things overlap, how collocated sampling locations are, etc. I would really encourage

the authors to come up with a better way to organize that information. Maybe a diagram would be helpful, one for summer and one for winter.

Page 8, line 27-28: you need to define what deviation in the upward or downward direction mean for CV. Is .5 better or worse than .1?

Page 9, line 9: The however doesn't seem to below here. This isn't really a contradiction of the previous statements.

Page 10, line 5: no comma here.

Page 10, line 9: Could these observations be coupled, e.g. clean high winds from the west lower local HONO concentrations to a level that the instruments have a difficult time measuring in resulting in poorer agreement?

Page 11, line 25-26: delete "for and extended single continuous period,"

Page 11, 29-30: "This suggests the YORK instrument was measuring HONO at reasonable concentrations." What does this even mean? Do you mean to say it is accurate? Precise? How do you determine which measurement is correct to evaluate the performance of the YORK instrument?

Page 12, line 30-31: The two-channel stripping coil will only perform well if the chemical interferant is not efficiently removed in the first stripping coil. If the interference is removed efficiently in the first coil the second coil experiences a significantly lower signal and therefore does not effectively remove the signal observed in the first coil. Please consider this possibility that the two coil system is not perfect at interference removal.

Page 12, line 33: I think you are trying to make the argument here that HONO should not partition to particles because they are acidic? Do you have evidence for that at the sampling location? What about other forms of particle nitrogen that could potentially yield signal in the instrument? You say do you have any measurements to back up any of the statements that you are making here?

Page 13, line 11-13: If water lowers the base line measurement compared to zero air this would impact the resulting background corrected ambient measurement. The percentage of ambient that is subtracted would large at low ambient HONO levels but would be negligible at large HONO levels. This could lead to large disagreement at low HONO levels. Yes, the calibration curve measured will only change in intercept, but the ambient data reported would be impacted.

———————————————————

---

## Author Comment (AC1) · 9 Oct 2019

**Response to reviewers' comments**

We thank the reviewers for their considered comments on our manuscript. Please find below our responses to their comments below.

**Reviewer 1**

This study reported an inter-comparison exercise between HONO measurements performed by multiple instruments. The results show that despite of good agreement on the temporal trends, the wet chemical methods consistently higher than the BBCEAS systems by between 12 and 39%. The reason for the divergence was not clear and the authors have speculated and discussed the potential influence of instrument locations. HONO is one of the most important precursors of OH radicals. Reliable measurements of HONO are key to understand its origin and role in the atmospheric chemistry. This study provides a great dataset to examine the performance and potential problems of HONO detectors in an atmosphere subject to strong anthropogenic influences. Overall, I think it is a nice study and would recommend its publication with revisions.

***Response:***

We thank the reviewer for the positive comments.

My main concerns are that (1) the authors may put too much weight on the contribution/discussion of spatial heterogeneity. Do you think that you may see a reverse relation (wet chemical concentration < BBCEAS) if you exchange the sampling locations?

***Response:***

This is difficult to answer definitively beyond speculatively. We do not know if we would see a reverse relation if the sampling locations were changed because we could not identify the mechanism/process that was driving the apparent disagreement between instruments. We have reduced the emphasis on the spatial heterogeneity, please see our response to reviewer 2, comment 1.

(2) the discussion about the daytime difference seems to be missing or mixed in the discussion. The 12% to 39% may not reflect the real difference up to several folds during daytime, the most relevant periods for the photochemical reactions.

***Response:***

While lower HONO concentrations are typically expected during the day due to photolysis this was not always the case in Beijing due to the level of haze attenuating solar UV (Shi et al. 2019). The agreement was between all instruments was highly linear ($r^2 > 0.97$), which indicates that the relationship between instruments was similar at all times. Overall, we found that the agreement appeared to be more dependent on the concentration (as also shown by the normalized difference analysis) irrespective of the time of day. To demonstrate the high degree of linearity between instruments, which do not show any sub-populations indicating (for example) different behavior between daytime and nighttime, we have added the scatter plots (shown below, Fig S4) to the supporting information.

[Figure]

**Figure S4**: Regression relationships of HONO measured by different instruments from the formal winter inter-comparison period (10 – 14 Nov 2016) at IAP, Beijing.  The blue line is the RMA regression and the black dashed line the 1:1 relationship.

Please find detailed comments below.

Page 1 line 36 "with the wet chemical methods consistently higher than the BBCEAS systems by between 12 and 39%." Is it the case for daytime, night time, I saw 100%?

*Response:*

We did not see any evidence for changes in agreement between instruments during the day, please see our previous response. While there were periods when the agreement changed, the 12-39% was range of differences observed for the overall regression analysis between the wet chemical and BBCEAS instruments during the winter intercomparison. To clarify this, we have changed the text to read:

*"with the wet chemical methods consistently higher overall than the BBCEAS systems by between 12 and 39%."*

Page 1 line 38 "The causes of the divergence in absolute HONO concentrations were unclear, and may in part have been due to spatial variability, i.e. differences in instrument location / inlet position." Did you check the concentration of other trace gases aerosols, T, RH, etc?

***Response:***

For the field campaign, there were unfortunately only multiple instruments/techniques for measuring HONO at the IAP site. In any case, other species like NOx and PM typically have much longer lifetime compared to HONO (in the order of hours/days compared to tens of minutes for HONO) and therefore we would not expect there to be significant variability for these species on the same spatial scales as HONO (Crilley et al. 2016).

Page 2 Introduction "contributing up to 40% of the OH budget in London (Lee et al., 2016)." The authors may consider referring to its contribution to the primary production of OH, where you can find more references (e.g., Kleffmann et al., GRL, 2005; Acker et al., GRL, 2006; Su et al., JGR, 2008) including measurements in Beijing (Yang et al., 2014).

***Response:***

We have added additional references on the contribution of HONO to OH budget for urban and rural areas:

*"The contribution of HONO photolysis to the OH budget can be significant in megacities, up to 33% in Beijing (Yang et al., 2014) and 40% in London (Lee et al. 2016) as well as forest (33%, Kleffmann et al. 2005) and rural areas (42%, Acker et al. 2006)."*

Page 2 Introduction "There are a number of known sources of HONO including direct emissions, heterogeneous reactions, homogenous, gas-phase reactions, biological processes and surface photolysis (see reviews by (Kleffmann, 2007;Spataro and Ianniello, 2014))." When talking about different sources, I'd suggest crediting the original research rather than exclusive referring to reviews.

***Response:***

There are many sources of HONO identified in the literature and to credit all the appropriate original research would add significant amount of text to the introduction. In the interests of brevity, we prefer to use the review articles, but have expanded these to include one further (earlier) study, and two significant papers published subsequent to the most recent available review. The text now reads:

*"There are a number of known sources of HONO including direct emissions, heterogeneous reactions, homogenous, gas-phase reactions, biological processes and surface photolysis (see reviews by (Lammel and Cape 1996;Kleffmann, 2007;Spataro and Ianniello, 2014) and recently abiotic and biotic processes on soils and biocrusts (Weber et al. 2015;Kim and Or 2019). "*

Page 2 line 14 "Positive artefacts can occur in inlet lines, as HONO is easily formed through heterogeneous reactions on wet surfaces (Zhou et al., 2003)." The study of Zhou is not about the heterogeneous reactions on inlets. It is discussing the formation of HONO and NOx from

photolysis of HNO3. Concerning the artefacts in inlet lines, you could refer to the studies of Zhou et al. (GRL,2002) and Su et al. (AE, 2008). The LOPAP style instruments have an outdoor sampling unit without additional inlet. Is this the case for the two wet chemical analyzers used here? What's the length of inlet of the other unspecified instruments?

***Response:***

We have corrected the reference for positive artefacts in inlet lines to Zhou et al. (2002) and Su et al (2008).

The BHAM LOPAP instrument used an outdoor sampling unit without additional inlet lines, precisely to avoid this issue. The ICCAS wet chemical instrument had the same inlet configuration as the BHAM instrument, and this information has been added to end of Section 2.2.2:

*"The ICCAS and BHAM instruments both used a similar outdoor sampling unit that employed a short quartz inlet (<2.5cm)."*

The CAM instrument inlet line length was 3 m long and this information has been added to Section 2.2.3:

*"The inlet line was ¼' outer diameter PFA tubing and was approximately 3m long."*

The AIOFM instrument inlet line was 4 m and this information has been added to Section 2.2.4:

*"The inlet line was ¼' outer diameter PFA tubing and was approximately 4m long."*

Page 8 line 26 "The co-efficient of variance (CV) is defined as the standard deviation divided by the mean and is used to compare the relative degree of variation between datasets." The authors should explain how to calculate CV explicitly because the mean and deviation can be calculated for an individual dataset. My understanding is that here you calculate CV of data from different instruments at each time step and it is better to clarify this

***Response:***

Yes, the reviewer is correct. We calculated the CV for each time step. To clarify the calculation of the CV, we have updated the text at the start of Section 3.1.2 to read:

*"We calculated the co-efficient of variance (CV) as a measure of the precision between the four instruments as per Eqn 1:*

$$CV = \frac{\sigma}{\mu}$$

*where $\mu$ is the mean and $\sigma$ the standard deviation for the measurements by all four instruments at a given 5 min interval. The CV was used to compare the relative degree of variation between datasets and as a guide a CV of 0.1 is considered as acceptable by the US EPA for PM instruments"*

Page 9 line 5 "the applied reference NO2 spectrum might contain absorption signatures from HONO. This would result in a higher NO2/HONO ratio retrieved from the BBCEAS compared to ambient air, and consequently reporting a low HONO mixing ratio (Kleffmann et al., 2006). However, the Voigt et al (2002) NO2 cross-section, used by both BBCEAS instruments, has previously been shown to have negligible HONO absorption structures (Veitel, 2002)." This part of discussion is not optimal. On one hand, you stated that the applied reference spectrum might be problematic. On the other hand, you referred to Veitel et al. 2002 saying that it is negligible. Actually, this is also the information provided by Kleffmann et al. (2006). Since you were using the one from Voigt et al., it should solve the impurity issue as suggested by Kleffmann et al. (2006). I'd suggest a reformulation of this part.

*Response:*

We have re-worded this sentence to read:

*"Both BBCEAS instruments use the Voigt et al (2002) NO₂ cross-section which has previously been shown to have negligible HONO absorption structures (Veitel, 2002; Kleffmann et al. 2006)".*

Page 9 line 15: "The systematic error for each instrument can be calculated by normalised sequential difference (NSD) according to Eqn 1 (Arnold et al., 2007)" The authors should better explain equation 1. What's the meaning of t and t+1, is it time? You later used i, j, what's their difference? Why it reflects a systematic error? People may have different ways to define a systematic error vs a random error. But the fact that BHAM was always higher than CAMB for me is a clear evidence for systematic errors.

*Response:*

To clarify the differences between the NSD and ND as well as their calculation, we have amended the text as follows:

*"Firstly, the systematic error for each instrument was calculated by normalised sequential difference (NSD) according to Eqn 1 (Arnold et al., 2007).*

$$NSD = \frac{(Conc_t - Conc_{t+1})}{(Conc_t \times Conc_{t+1})^{0.5}} \qquad (1)$$

*NSD is a method of calculating the variation between consecutive measurements for an individual instrument, where $Conc_t$ is the concentration measured at time t and $Conc_{t+1}$ the following measurement. The results are shown in Fig S1 (Supporting Information), and as each instrument showed a symmetrical and Gaussian distribution it suggests there was no internal systematic bias for any given instrument.*

*Secondly, we then examined the normalized difference (ND) between pairs of instruments to explore inter-instrument variability, calculated according to Eqn 2 (Pinto et al., 2014):*

$$ND_{ij} = \frac{(C_i - C_j)}{(C_i + C_j)} \qquad (2)$$

*where $C_i$ and $C_j$ denote HONO levels measured by any pair of instruments (BHAM, ICCAS, AIOFM or CAM), calculated for each measurement period. For example, the ND for the BHAM and CAM instruments ($ND_{BHAM-CAM}$) would be calculated by ($[HONO]_{BHAM}$ - $[HONO]_{CAM}$)/ ($[HONO]_{BHAM}$ + $[HONO]_{CAM}$)."*

Page 10 line 2 "The ND was evaluated as function of wind direction and measured HONO concentration (Fig 3), to explore if ambient concentration or spatial heterogeneity could explain the disagreements." Do you have other parameters (e.g., SO2, O3, NOx, etc) that have been measured in different containers? A comparison of these parameters may give you a better idea about the spatial heterogeneity and help to disentangle it from the other effects. Besides, the spatial heterogeneity and HONO concentration might not be the only reasons for the observed differences. Artefacts and intrinsic limitation of individual methods could also results in different kinds of disagreement. These, however, were not discussed in this study. Do you consider these as minor issues here?

***Response:***

There were measurements of a range of gas and particle phase parameters during the field campaigns at the IAP site (Shi et al. 2019). We chose to focus on the wind direction and HONO concentration as these would indicate if was any influence from spatial heterogeneity in HONO levels (wind direction) or detection limits or instrument response (HONO concentration) on the measured levels by the four instruments. For other parameters such as ozone or $SO_2$ we would not expect these to have any influence on the measured HONO levels by any of the instruments (e.g. as an interferent, as demonstrated previously, Helend et al., 2001; Kleffmann et al., 2002) and so we did not consider them for this analysis. Furthermore, as we showed earlier in Section 3.1.2, the levels of $NO_2$ did not influence the measured concentrations for any instruments.

Page10, Section3.1.4. Here the authors investigated the concentration dependence of instrument agreement through comparison of slopes between the whole dataset and a subset of data. Can you give more details why you were expecting a concentration dependence, or what could lead to a concentration dependence? Do you think the detection limit would be an issue here? This can be easily checked by exclusively comparing data above the threshold.

***Response:***

The results from the CV (Fig 2) and ND (Fig 3) analyses suggested that the level of agreement between instruments decreased at low HONO levels (see section 3.1.4,). This is why we chose to investigate the concentration dependence, and the relationship between instruments at low HONO levels by linear regression. As noted in section 3.1.4, the only change in relationship observed was between CAM relative to the other instruments (BHAM, ICCAS and AIOFM), which we suggest may be related to differences in instrument sensitivity.

I also have some technical questions: (1) how did you conclude "the observed slopes between the BHAM-ICCAS-AIOFM at low concentrations (<2 ppb) were similar to those .."? Because according to the confidence interval in Table 2 and 4, these slopes could be significantly

different; (2) what's the criteria of choosing 2 ppb as the threshold, will the results change if you take other values, e.g., 0.5 ppb?

***Response:***

1. The reviewer rightly points out that the slopes between BHAM-ICCAS-AIOFM were not within the stated confidence intervals for low concentrations (<2 ppb, Table 4) compared to the whole dataset (Table 2). The point was not that the slopes were the same but that were similar. This contrasts with what was observed for the agreement with the CAM instrument to the other instrument, where the slopes were markedly decreased at low concentrations (Table 4) relative to that observed for the whole dataset (Table 2). We have changed the text in section 3.1.4 to read:

   *"From Table 4, the observed slopes between the BHAM-ICCAS-AIOFM at low concentrations (<2 ppb) were similar to those for the whole winter inter-comparison dataset (Table 2) unlike when compared to the CAM instrument. This suggests that the difference in measured concentrations between these instruments (BHAM-ICCAS-AIOFM, as indicated by the slope) was not related to concentration. The notable decrease observed in the slope for the low concentrations between CAM and the other three instruments compared to whole inter-comparison (Tables 2 and 4, respectively), potentially points to changes in the CAM readings at lower concentrations. This change may be related to differences in instrument sensitivity (Table 1)."*

2. We choose 2 ppb as a threshold to ensure there were enough data for the linear regression. If we use a threshold of 1.0 ppb, the absolute values of the slopes do change but the overall trends do not, that is we observe similar slopes between BHAM-ICCAS-AIOFM and lower slopes between the CAM instrument and the other three instruments.

A more straight forward way to investigate the concentration dependence would be directly plot the difference or normalized difference against the concentration. HONO concentration also has a strong diurnal cycle, with a minimum around noon time with active photochemistry and temperature. The concentration dependence of bias might also be caused by a diurnal cycle of the interference signals as discussed in Su et al. (AE 2008). Can you check if the difference in your study also has a prominent diurnal cycle?

***Response:***

When we plot the ND as function of concentration, there is increase in the ND at low concentrations for all instruments (shown below, Fig 1). This information is also shown in Fig 3 (where the ND is plotted as a function of wind direction coloured by concentration). As part of the discussion in Section 3.1.4, we mention the relationship between ND and concentration (page 10, line 4):

*"From Figure 3, for all instrument pairs the highest ND, and therefore largest relative difference, between instruments was at low HONO mixing ratios (ca. <1 ppb)"*

[Figure]

Figure 1: Normalised Differences (ND) for each instrument pair as a function of concentration as measured during the winter inter-comparison.

When we plot the diurnal cycle of the ND for each instrument pair (shown below, Fig 2), we do not see any noticeable diurnal trends for all instruments. For some instruments there is a peak in the afternoon, which may point to some photochemical influence but as this was not observed by all instrument pairs this may be more of a statistical artifact. We note that this shape is also apparent between the optical instrument pair, and between the wet chemical instrument pair. We also note that we only have a short time series and therefore any outliers may also have affected the results. As we have mentioned in previous responses, the overall ND and CV analyses indicate that HONO levels rather than time of day were the stronger determinant in affecting the instrument agreement.

[Figure]

Figure 2: Mean diurnal trends in normalized differences (nd) for each instrument pair during the formal winter inter-comparison. The shaded areas represent 95% confidence intervals.

**Reviewer 2**

Review of "Intercomparison of nitrous acid (HONO) measurement techniques in a megacity (Beijing)", AMT-2019-139 This paper presents a multi instrument comparison of HONO observations collected in Beijing. Considering the atmospheric importance of HONO, it is very important to determine the accuracy of various measurement techniques. The most refreshing part of this manuscript is that the authors make no attempts to disguise the differences between the various techniques, and do not attempt to identify a true measurement technique. This allows for a neutral consideration of the state of available technology to measure ambient HONO, while making it difficult to understand the cause of the variations. The presentation and discussion could use some work cleaning up and organizing the details of the results as it is sometimes confusing to keep track of what methods or time periods are being compared or can in fact be compared. I also find that the authors put a lot of stock in the location of sampling points as an explanation for the discrepancies, while providing limited evidence to support this. This work is valuable in that it draws attention to the continuing problem of evaluating the robustness of HONO observations. This work would benefit from attempting to arrive at a point where evidence-based suggestions can be provided to inform future observations or comparison studies. I support the publication of this work contingent upon addition minor revisions as suggested above and in the detailed comments provided below.

***Response:***

We thank the reviewer for their positive comments, especially regarding our transparency with the observed differences between instruments. We also would like to have arrived at more robust conclusions to the cause of the divergence between the different techniques but were unable to

definitively do so, within the available datasets. This may be because there were multiple factors that were driving the differences between instruments. Based upon the normalized difference analysis (ND, Section 3.1.3) the differences between instruments appeared to be related to HONO concentration, wind speed and direction. Thus, we believe this is evidence that spatial variability in HONO concentration may have affected the inter-comparison. As we discussed in the text, this may have been due to the horizontal spread of instrument inlets during the intercomparison (up to 13 m apart). However, we do acknowledge that the preceding discussion is not conclusive evidence for differences in inlet locations and hence spatial variability in HONO concentration affecting the intercomparison. It may be that the observed relationship between ND and wind direction/speed was more associative than casual.

Considering the above, we have toned down the conclusions regarding the effect of the inlet location, changing end of abstract to read (page 1, line 38):

*"The causes of the divergence in absolute HONO concentrations were unclear and may in part have been due to spatial variability, i.e. differences in instrument location / inlet position but this observation may have been more associative than casual."*

We have also added the discussion above on the spatial heterogeneity (starting at page 12, line 13):

*"However, in the current work the results do not conclusively point to spatial heterogeneity in HONO concentrations affecting the results. As both the current work and Pinto et al. (2014) found some evidence for spatial heterogeneity in HONO concentrations affecting their intercomparisons, this would suggest that to avoid this issue future studies should use a common inlet for all instruments in the field."*

Page 2, line 3-4: This is oddly specific towards London considering the number of publications on daytime HONO, and that this manuscript does not address London HONO. Suggest broadening the citations for the generalization of the introduction.

***Response:***

We have added additional citations to cover work elsewhere, including previous HONO work in China, please see our response to reviewer 1.

Page 2, line 9: The statement "depending on the proximity to emission sources" is meant to mean what? Emission of HONO, NOx? Direct, indirect?

***Response:***

We have changed this sentence to read:

*"high spatial heterogeneity in HONO concentration can be observed depending on the proximity to sources of direct emissions of HONO".*

Page 3, line 14: "highly polluted locations like Beijing" needs a citation or evidence otherwise that Beijing is highly polluted.

*Response:*

We have added the references Tong et al. 2016 and Wang et al 2017.

Page 4, line 3: edit to read "Measurements were performed as part of"?

*Response:*

We have made this change.

Page 4, line 15: "referred to as LOPAP throughout" is a bit confusing since you just introduced that instrument will be referred to by the institution and have already defined LOPAP acronym earlier in the manuscript.

*Response:*

We have edited this text to read:

*"The University of Birmingham operated a LOPAP (QUMA Elektronik & Analytik GmbH) at IAP."*

Page 4, line 25: It would be helpful in this work to explicitly discuss what the main differences between the two LOPAP techniques are.

*Response:*

There were two main differences between the BHAM and ICCAS system; 1) the method used for baseline correction and 2) the length of the optical path length. This information has bee added to the end of Section 2.2.2:

*"While the BHAM and ICCAS instruments operated according to the same principles, there were two main differences. The first was the method for determining the baseline, the BHAM instrument used an overflow of $N_2$ while the ICCAS instrument replaced the reagents with water. The second was the optical path length, which was 2.0 and 0.5 m for the BHAM and ICCAS instruments, respectively."*

Page 5, line 12: AIOFM is not yet defined in the text.

*Response:*

The text has been changed to read;

*"the instrument was moved to an adjacent container, also housing the other BBCEAS instrument"*

The acronym AIOFM is defined in the next section.

Page 54, line 16-21: This is pretty hard to understand here. Are you simply stating that the flow controller was not properly calibrated and describing the process to calibrate it? If so this seems like it is an unnecessary discussion as one would expect that you have calibrated your flow controllers to get accurate measurements.

*Response:*

Yes, we are describing the calibration of the flow controller that was revised after the measurements. We feel necessary to keep a complete record of how/when calibrations have been completed and applied for the flow controller.

Page 6, line 3-4: How exactly did you consider sample loss and secondary formation of HONO in the instrument? That is not exactly the most straightforward thing to do, and it is glossed over here.

*Response:*

A HONO standard generator was developed to supply stable concentrations of HONO and the experiment of sample loss for CEAS-AIOFM was operated in the laboratory as following: 1. HONO from the standard generator directly flowed into the IBBCEAS instrument, with its steady-state concentration measured by IBBCEAS. 2. The HONO source was replaced by a fast $N_2$ flow (10 SLPM) which was kept running for about 5 mins. 3. A second 1 μm PTFE filter and 3-m length PFA inlet tube and a piece of PFA tube of the same dimension as that of the optical cavity were added upstream of the IBBCEAS cavity to reproduce any potential loss on the particle filter, inlet and cavity walls. The HONO flow was re-introduced through the extra components and the IBBCEAS cavity and the new steady-state HONO concentration was measured by IBBCEAS. 4. The particle filter, PFA inlet tube, and the PFA "cavity" tube were all removed, and pure nitrogen was again flowed through the IBBCEAS instrument. In this experimental cycle, the relative humidity (RH) was about 65 % and temperature was about 23 °C, the sample loss of the IBBCEAS instrument for HONO was found to be about 2.0 % (from average 46.0 ppb to average 45.1 ppb). We also repeated this experiment at different RH levels and found that the sample loss of the IBBCEAS instrument for HONO was about 2.1 % at 25 % RH and about 1.9 % at 50 % RH, suggesting a weak RH dependence of the sample loss of the CEAS instrument for HONO.

To investigate any potential secondary HONO formation for CEAS-AIOFM on the inlet or cavity walls from $NO_2$, about 80 ppb $NO_2$ at different RH levels (about 20% RH, 30% RH, 50% RH and 70% RH) flowed through a 3-m PFA inlet tube into the IBBCEAS instrument for a long time at typical sampling flow rates, respectively, no [HONO] was observed in the optical cavity, suggesting that the secondary HONO formation is negligible for this IBBCEAS instrument under this typical operation condition.

Page 6, line 24 and elsewhere: One major thing missing here that is likely very important is a detailed discussion on how the function instrument zero was determined and the associated errors involved. For example, how was the absolute error of 33ppt calculated here? Is that the variability between instrument zeros? Is once every 45 minutes sufficient to capture the variability in the background for the CIMS. Are the timescales for others important? From figure 1 it seems like incorrect background subtraction for each instrument could easily explain the difference in magnitude between the various instruments, especially considering the high degree of correlation observed. I'd really like to see a more detailed discussion of the background determination and subtraction and its impact on the reported values for each instrument.

*Response:*

For the CIMS, we agree that that background determination is an important omission that was also raised elsewhere. We have therefore, on your recommendation included figures and more detailed text to show this process over an example time period. The variability between backgrounds is low and therefore we feel that once every 45 minutes in sufficient to capture the variability, particularly as the instrument detects a range of species and it is infeasible to do species specific backgrounds.

We have now included a figure which shows the variation in background over a range of typical sample and background measurement cycles as well as the following text to describe the background determination.

*"The CIMS instrument zero was determined by flowing dry nitrogen into the IMR periodically, and the backgrounds were applied consecutively. As shown in Figure S1, there was very little variability of this background during the measurement period. Though the overflowing of dry N2 will have an effect on the sensitivity of the instrument to those compounds whose detection is water dependent, due to the low instrument backgrounds, the absolute error remains small and we deem this an acceptable limitation in order to measure a vast suite of different compounds for which no best practice backgrounding method has been established. We therefore calculated the absolute error of 33 ppt as 3 sigma deviations of the background signal."*

[Figure]

Figure S1: Illustration of the backgrounding procedure used in the CIMS instrument

We have also added further discussion on the effect of the different background corrections for the BHAM and ICCAS wet chemical instruments, please see our response to the final comment.

Page 6, line 30: The lack of humidity in the dry N2 zero should significantly impact the HONO sensitivity, you should include the humidity dependent calibration of your instrument. Does HCOOH display the same dependence? This could impact your ability to use formic acid as a surrogate to track changes in instrumental sensitivity. While this is commonly done in past studies it may not be accurate. This is also a concern for the SIFT instrument backgrounds.

*Response:*

For the CIMS, we agree that this method of backgrounding will have associated errors, however with an instrument of this type specific backgrounding for individual species is not feasible (also discussed in previous response in more detail). Furthermore, the formic acid calibrations were all carried out under the same conditions meaning that they are comparable to one another and can therefore be used as a valid surrogate to track changes in instrumental sensitivity.

In terms of the SIFT instrument background, Spanel and Smith (2000) investigated the detection of gaseous HONO using SIFT-MS. This was done through the analysis of a sample of acidified nitrite headspace. Bimolecular reaction of $H_3O^+$ and nitrous acid produces $H_2NO_2$ (m/z 48, 67%) and $NO^+$ (m/z 30, 33%). The rate constant (k) of this exothermic proton transfer reaction is calculated to be 2.7 x $10^{-9}$ $cm^3s^{-1}$ with respect to hydronium ($H_3O^+$) and 2.2 x $10^{-9}$ $cm^3s^{-1}$ with respect to hydronium mono-hydrate $(H_3O \cdot H_2O)^+$ (Spanel and Smith, 2000). Nitrous acid does not undergo proton transfer with hydronium di-hydrate $(H_3O \cdot H_2O^2)^+$ and tri-hydrate $(H_3O \cdot H_2O^3)^+$ in SIFT-MS. Nitrous acid mixing ratios herein were determined using the branching ratio corrected protonated product ion m/z 48 intensity normalized to both $H_3O^+$ and $H_3O \cdot H_2O$ with their respective k values. As such, calculated HONO mixing ratios using SIFT-MS should be independent of the humidity of the gas sample. This description has been added to the method section of the manuscript for clarification. However, we do recognize that humidity dependent surface effects are not taken into account via this reagent ion normalization. We had a platinum catalytic converter (held at 380°C) based zero air generator for routine background determination of a wide range of VOCs (not covered in this paper) using the SIFT-MS. Unfortunately, HONO was appreciably higher in this background, likely due to the thermal decomposition of alkyl amides, and so could not be used for HONO background subtraction. Dry nitrogen based backgrounds were the only available alternative. Hourly HONO gas phase backgrounds in nitrogen were 110±40 pptv during the measurement period presented and as such are unlikely to have a significant contribution on the determined mixing ratio.

Page 7, line 20: The compounds used in the relative transmission calibration curve are very difference from HONO. How well is method expected to perform in approximating the HONO sensitivity considering these differences? Are the portions of a given molecules real sensitivity that are not accurately captured by this method, for example surface losses or secondary ion chemistry, or fragmentation?

*Response:*

For data herein, the secondary ion chemistry and fragmentation of HONO using SIFT-MS is well documented and has been taken into account for the data presented. The bimolecular reaction of $H_3O^+$ and nitrous acid produces $H_2NO_2$ (m/z 48, 67%) and $NO^+$ (m/z 30, 33%). No other fragment ions occur. The rate constant (k) of this exothermic proton transfer reaction is calculated to be 2.7 x $10^{-9}$ $cm^3s^{-1}$ with respect to hydronium ($H_3O^+$) and 2.2 x $10^{-9}$ $cm^3s^{-1}$ with respect to hydronium mono-hydrate $(H_3O \cdot H_2O)^+$ (Spanel and Smith, 2000). Nitrous acid does not undergo proton transfer with hydronium di-hydrate $(H_3O \cdot H_2O^2)^+$ and tri-hydrate $(H_3O \cdot H_2O^3)^+$ in SIFT-MS (Spanel and Smith, 2000). Nitrous acid mixing ratios herein were determined using the branching ratio corrected protonated product ion m/z 48 intensity normalized to both $H_3O^+$

and $H_3O \cdot H_2O$ with their respective k values. As such, calculated HONO mixing ratios using SIFT-MS should be independent of the humidity of the gas sample.

A known sample artefact is production of HONO on internal surfaces exposed to $NO_2$ and $H_2O$. Previous studies (Muller, et al 2016) involving PTR-MS have investigated this artefact through the introduction of a gas-phase $NO_2$ standard to a humidified inlet stream. In this case, a high concentration $NO_2$ stream (25 ppbv) produced a signal corresponding to 3.5% of what the same concentration of HONO would produce. Considering the literature, it is generally suggested that a fast, turbulent sample flow is recommended during sampling which minimises sample residence time within sample lines in order to reduce this effect. These considerations were implemented during our sampling procedure.

As stated, a HONO calibration source was not available for on-site calibration of the SIFT-MS, hence mixing ratios were calculated using the instrument specific transmission coefficients and reaction rates taken from Spanel and Smith, 2000. The instrument specific transmission coefficients were calculated daily. Despite the careful calculation of coefficients, previous studies (de Gouw and Warneke, 2007) have suggested mixing ratios calculated using this approach can have large systematic errors, therefore some systematic bias in mixing ratios cannot be ruled out here and are difficult to quantify. Future work will be to develop a portable HONO gas generation device to directly calibrate SIFT-MS in the field and reduce this bias.

Page 7, line 24-25: This is a poorly written sentence and needs work.

***Response:***

This sentence has been edited to read:

*"The formal inter-comparison of the four established techniques for measuring HONO (2 wet chemical and 2 BBCEAS) took place during the 9th-14th November 2016."*

Page 8, line 1-2: It is very difficult throughout this manuscript to keep track of when things overlap, how collocated sampling locations are, etc. I would really encourage the authors to come up with a better way to organize that information. Maybe a diagram would be helpful, one for summer and one for winter.

***Response:***

We have added a diagram of the winter and summer inlet locations to the SI to clarify the inlet locations for all instruments, shown below for winter.

[Figure]

Figure S2: Schematic indicating the relative position of each instrument inlet during the winter inter-comparison. Each rectangle represents a shipping container laboratory. Note not to scale.

Page 8, line 27-28: you need to define what deviation in the upward or downward direction mean for CV. Is .5 better or worse than .1?

*Response:*

An increase in the CV indicates that the agreement between the instruments decreased. To clarify this point, we have altered the text to read:

*"The CV was however observed to increase at the end of the inter-comparison, coinciding with period of the lowest mean HONO concentration (< 1 ppb, Fig 2). An increase in the CV indicates worsening agreement between instruments possibly due to the concentrations approaching the detection limit (DL) of some instruments (Table 1)."*

Page 9, line 9: The however doesn't seem to below here. This isn't really a contradiction of the previous statements.

*Response:*

Sentence now reads:

*"Overall, Fig 2 demonstrates no apparent relationship between the CV and $NO_2$."*

Page 10, line 5: no comma here.

*Response:*

Fixed

Page 10, line 9: Could these observations be coupled, e.g. clean high winds from the west lower local HONO concentrations to a level that the instruments have a difficult time measuring in resulting in poorer agreement?

*Response:*

These observations could be coupled, but as we do not have an explanation as to why there could be low HONO concentrations associated with westerly winds, given the reasonably homogeneous surrounding areas, we do not wish to speculate in the manusript.

Page 11, line 25-26: delete "for and extended single continuous period,"

*Response:*

Done.

Page 11, 29-30: "This suggests the YORK instrument was measuring HONO at reasonable concentrations." What does this even mean? Do you mean to say it is accurate? Precise? How do you determine which measurement is correct to evaluate the performance of the YORK instrument?

*Response:*

The reviewer makes a valid point and we have removed this sentence from the text.

Page12, line30-31: The two-channel stripping coil will only perform well if the chemical interferant is not efficiently removed in the first stripping coil. If the interference is removed efficiently in the first coil the second coil experiences a significantly lower signal and therefore does not effectively remove the signal observed in the first coil. Please consider this possibility that the two coil system is not perfect at interference removal.

*Response:*

The reviewer rightly points out that the two-channel stripping coil will not effectively account for any interference if the chemical interferent is removed completely in the first coil. It is important to note that the azo-dye will only react with nitrite in solution. The work by Heland et al. (2001) and Kleffmann et al. (2002) conclusively showed that for known chemical interferents that may react in solution with the azo-dye (e.g. $NO_2$ and PAN) the real interference (Ch1-Ch2) was small, in order of 0.01%. We of course cannot discount that an unknown chemical interferent being sampled completely in coil 1 but this has been shown to be unlikely to be significant based upon results for known compounds (e.g. $NO_2$).

Page 12, line 33: I think you are trying to make the argument here that HONO should not partition to particles because they are acidic? Do you have evidence for that at the sampling location? What about other forms of particle nitrogen that could potentially yield signal in the instrument? You say do you have any measurements to back up any of the statements that you are making here?

***Response:***

We are making the argument that due to the acidic nature of the aerosol in Beijing (Song et al. 18) we do not expect there to be much particle-phase nitrite. Furthermore, Broske et al. (2003, from Kleffmann 2006) demonstrated that the uptake of aerosol in the 50-800 nm fraction by the LOPAP sampling inlet is small (in the order of 1%). While larger particles may be sampled by impaction, Kleffmann et al. (2006) argued that this unlikely to be significant as the uptake would need to be an order of magnitude higher (>10%) for it to be significant interferent, which seems unlikely. Consequently, we do not expect there to be significant signal from other forms of particle nitrogen. To clarify our arguments, we have modified the text to read:

*"The aerosol in Beijing is typically acidic (Song et al. 2018) and based on the effective Henry's law constant for HONO we would expect there to be little particle-phase nitrite (Kleffmann et al. 2006). This combined with the expected low uptake of particles by the LOPAP sampling inlet (in order of 1% for particles with a diameter between 50-800 nm, Broske et al. 2003) suggests that there would be limited chemical interference from particle-phase species. We also note that particle-phase chemical interference would likely be corrected for by the two-channel system."*

Page 13, line 11-13: If water lowers the base line measurement compared to zero air this would impact the resulting background corrected ambient measurement. The percentage of ambient that is subtracted would large at low ambient HONO levels but would be negligible at large HONO levels. This could lead to large disagreement at low HONO levels. Yes, the calibration curve measured will only change in intercept, but the ambient data reported would be impacted.

***Response:***

We investigated the agreement between BHAM and ICCAS instruments at low concentrations (<2ppb) in Section 3.1.4 and found that the relationship between BHAM and ICCAS instruments was similar to that observed over the whole measured range. The reviewer is right to point out that the change in baseline measurements would have more impact at lower concentrations and we have added the following text to clarify this point at page 13, line 11:

*"Tests have shown that water results in a lower baseline measurement for the LOPAP (approx. 80-100 ppt). We note that the ambient HONO was typically within the ppb range during the inter-comparison (Fig 1) and the effect of this baseline difference would be negligible at these levels. But at lower concentrations (low ppt), it would proportionally have a greater influence on the reported HONO levels by the BHAM and ICCAS instruments. High ND was observed at low concentration (<1ppb, Fig 3) and the difference in absolute baseline correction may explain this."*

This paper presents measurements of HONO by several instruments and is of interest to many researchers (myself included) who are interested in HONO. Given the growth in the use of the iodide ToF chemical ionization mass spectrometer method, it would be helpful for further details of the instrument's operation to be presented. The IMR pressure is held at 400 mbar, which is higher than that used for most research groups' IMRs (∼50 – 100 mbar). What is the reason for operating at this relatively high pressure?

*Response:*

We agree that this is higher than normal, as a group we also run our IMR pressure at 50-100 mbar when using the Po-210 source, as is recommended by the instrument supplier and from our detailed experimental understanding of the instrument. In China the use of Po-210 as is normal, was not possible, and we therefore used the Tofwerk X-ray ionisation source within the range of operating pressures suggested by the instrument supplier (300-500 mbar). This is also in line with conditions which have been used in previous studies (Breton et al. 2018). We have added the following text to the method to clarify this:

*"This flow enters an ion molecule region (IMR) which was maintained at a pressure of 400 mbar using an SSH-112 pump fitted with an Aerodyne pressure control box to account for changes in ambient pressure. The IMR pressure is significant higher than is usual for this CIMS instrument when using PO-210, but is necessary given the change in ionisation source in this study. Operation is comparable to the Breton et al., (2018) study who also used the same Tofwerk X-ray ionisation source."*

Backgrounds were determined using dry N2. The authors state "The overflowing of dry N2 will have a small effect on the sensitivity of the instrument for those compounds C1 whose detection is water dependent. Here we find that due to the very low instrumental background for HONO, the absolute error remains small (the absolute error remains small (<33 ppt)". According to the same Lee et al. 2014 paper referenced in the manuscript, the sensitivity (as configured in Lee et al) varies by a factor of five between dry conditions and the most humid conditions tested (PH2O = 0.8 mbar). It would be useful if there were a figure that showed the time series for m/z 174 for a typical sample & background measurement cycle to support the statement that the uncertainty in the background is essentially inconsequential.

*Response:*

We have now included a figure which shows the variation in background over a range of typical sample and background measurement cycles as well as additional text to describe the background correction method. Please see our response to Reviewer 1.

The sensitivity is quoted as 0.28 counts/s/ppt – at what humidity?

*Response:*

This sensitivity was reported at 5% relative humidity.

A graph of the HONO sensitivity as a function of humidity would be helpful, especially since it could differ than that measured by Lee et al given the different IMR pressures used. Is this sensitivity normalized to 1,000,000 cps of reagent ion? What is the total reagent ion signal?

***Response:***

This wasn't done, and this is now noted in the manuscripts as a limitation of this method. This was not deemed necessary to do because there was little variation in the I.H2O:I- ratio throughout the day. The sensitivity is not reported normalized to 1,000,000 cps as with X-Ray ionisation CIMS operates at much lower counts of ca. 100,000 and therefore we feel that normalisation of sensitivity to 1 million does not represent the actual data. A critical parameter which determines sensitivity which enables sensitivities to be more widely compared, however is not widely reported in literature is the ToF extraction frequency. Our calibrations were carried out at a ToF extraction frequency of 16 kHz. The following text has been added to the method section to clarify:

*"A limitation of the CIMS calibration approach for HONO is that it was not established as a function of humidity. This was not deemed necessary because there was an average variation of only 2% in the I-:IH2O- ratio throughout the day."*

**References**

Acker K, Möller D, Wieprecht W, Meixner FX, Bohn B, Gilge S, Plass-Dülmer C, Berresheim H. Strong daytime production of OH from HNO2 at a rural mountain site. Geophysical Research Letters. 2006 Jan;33(2).

Bröske R, Kleffmann J, Wiesen P. Heterogeneous conversion of NO 2 on secondary organic aerosol surfaces: A possible source of nitrous acid (HONO) in the atmosphere?. Atmospheric Chemistry and Physics. 2003 May 13;3(3):469-74.

Breton ML, Wang Y, Hallquist ÅM, Pathak RK, Zheng J, Yang Y, Shang D, Glasius M, Bannan TJ, Liu Q, Chan CK. Online gas-and particle-phase measurements of organosulfates, organosulfonates and nitrooxy organosulfates in Beijing utilizing a FIGAERO ToF-CIMS. Atmospheric Chemistry and Physics. 2018 Jul 19;18(14):10355-71.

Crilley, L. R., Kramer, L., Pope, F. D., Whalley, L. K., Cryer, D. R., Heard, D. E., Lee, J. D., Reed, C., and Bloss, W. J.: On the interpretation of in situ HONO observations via photochemical steady state, Faraday Discussions, 189, 191-212, 10.1039/C5FD00224A, 2016.

de Gouw, J. and Warneke, C.: Measurements of volatile organic compounds in the Earth's atmosphere using proton-transferreaction mass spectrometry, Mass Spectrom. Rev., 26, 223–257, 2007.

Kleffmann J, Gavriloaiei T, Hofzumahaus A, Holland F, Koppmann R, Rupp L, Schlosser E, Siese M, Wahner A. Daytime formation of nitrous acid: A major source of OH radicals in a forest. Geophysical Research Letters. 2005 Mar;32(5).

Kleffmann J, Lörzer JC, Wiesen P, Kern C, Trick S, Volkamer R, Rodenas M, Wirtz K. Intercomparison of the DOAS and LOPAP techniques for the detection of nitrous acid (HONO). Atmospheric Environment. 2006 Jun 1;40(20):3640-52.

Kim M, Or D. Microscale pH variations during drying of soils and desert biocrusts affect HONO and NH 3 emissions. Nature communications. 2019 Sep 2;10(1):1-2.

Lammel G, Cape J. Nitrous acid and nitrite in the atmosphere. Chemical Society Reviews. 1996;25(5):361-9.

Muller, M.; Anderson, B.; Beyersdorf, A.; Crawford, J. H.; Diskin, G.; Eichler, P.; Fried, A.; Keutsch, F. N.; Mikoviny, T.; Thornhill, K. L.; Walega, J. G.; Weinheimer, A. J.; Yang, M.; Yokelson, Robert; and Wisthaler, A., "In situ measurements and modeling of reactive trace gases in a small biomass burning plume" (2015). Chemistry and Biochemistry Faculty Publications. 92. https://scholarworks.umt.edu/chem_pubs/92

Pinto JP, Dibb J, Lee BH, Rappenglück B, Wood EC, Levy M, Zhang RY, Lefer B, Ren XR, Stutz J, Tsai C. Intercomparison of field measurements of nitrous acid (HONO) during the SHARP campaign. Journal of Geophysical Research: Atmospheres. 2014 May 16;119(9):5583-601.

Shi Z, Vu T, Kotthaus S, Harrison RM, Grimmond S, Yue S, Zhu T, Lee J, Han Y, Demuzere M, Dunmore RE, et al. Introduction to the special issue "In-depth study of air pollution sources and processes within Beijing and its surrounding region (APHH-Beijing)". Atmospheric Chemistry and Physics. 2019 Jun 5;19(11):7519-46.

Spanel, P. Smith, D. An investigation of the reaction of $H_3O^+$ and $O_2^+$ with NO, $NO_2$, $N_2O$ and $HNO_2$ in support of selected ion flow tube mass spectrometry. *Rapid Comms in Mass Spectrometry, 14, 8, 2000*

Song S, Gao M, Xu W, Shao J, Shi G, Wang S, Wang Y, Sun Y, McElroy MB. Fine-particle pH for Beijing winter haze as inferred from different thermodynamic equilibrium models. Atmospheric Chemistry and Physics. 2018 May 28;18(10):7423-38.

Tong S, Hou S, Zhang Y, Chu B, Liu Y, He H, Zhao P, Ge M. Exploring the nitrous acid (HONO) formation mechanism in winter Beijing: direct emissions and heterogeneous production in urban and suburban areas. Faraday discussions. 2016 Jul 14;189:213-30.

Wang, J., Zhang, X., Guo, J., Wang, Z., and Zhang, M.: Observation of nitrous acid (HONO) in Beijing, China: Seasonal variation, nocturnal formation and daytime budget, Science of The Total Environment, 587-588, 350-359, https://doi.org/10.1016/j.scitotenv.2017.02.159, 2017.

Weber B, Wu D, Tamm A, Ruckteschler N, Rodriguez-Caballero E, Steinkamp J, Meusel H, Elbert W, Behrendt T, Soergel M, Cheng Y. Biological soil crusts accelerate the nitrogen cycle through large NO and HONO emissions in drylands. Proceedings of the National Academy of Sciences. 2015 Dec 15;112(50):15384-9.

Yang Q, Su H, Li X, Cheng Y, Lu K, Cheng P, Gu J, Guo S, Hu M, Zeng L, Zhu T. Daytime HONO formation in the suburban area of the megacity Beijing, China. Science China Chemistry. 2014 Jul 1;57(7):1032-42.